# Estimating Riemannian Metric with Noise-Contaminated Intrinsic Distance

**Jiaming Qiu**
Fred Hutchinson Cancer Center
`jqiu3@fredhutch.org`

**Xiongtao Dai**
Division of Biostatistics, School of Public Health
University of California, Berkeley
`xdai@berkeley.edu`

## Abstract

We extend metric learning by studying the Riemannian manifold structure of the underlying data space induced by similarity measures between data points. The key quantity of interest here is the Riemannian metric, which characterizes the Riemannian geometry and defines straight lines and derivatives on the manifold. Being able to estimate the Riemannian metric allows us to gain insights into the underlying manifold and compute geometric features such as the geodesic curves. We model the observed similarity measures as noisy responses generated from a function of the intrinsic geodesic distance between data points. A new local regression approach is proposed to learn the Riemannian metric tensor and its derivatives based on a Taylor expansion for the squared geodesic distances, accommodating different types of data such as continuous, binary, or comparative responses. We develop theoretical foundation for our method by deriving the rates of convergence for the asymptotic bias and variance of the estimated metric tensor. The proposed method is shown to be versatile in simulation studies and real data applications involving taxi trip time in New York City and MNIST digits.

## 1 Introduction

With increasing data complexity, it has received increasing attention for capturing the geometric structure of the data space characterized by a Riemannian manifold, which consists of two vital parts: the coordinate system and the Riemannian metric. Manifold learning attempts to find low-dimensional representations as coordinates for the high-dimensional data [23, 27, 7]. Yet existing methods generally assume the local geometry is given, e.g., by the Euclidean distance between ambient data points, leading to non-isometric methods that distort the data geometry in the embedded space. On the other hand, many methods have been proposed to consider the Riemannian metric, including parametric estimation [17, 16, 15]; multi-metric learning [12]; pushforward via the Laplacian operator [21]; pull-back via Jacobian of generative models [1, 2]; Bayesian approach based on arc length of Gaussian process [14]; and dynamics of evolving probability when time-indexed samples are available [24]. It is also possible to recover the manifold in an abstract setting with only similarity measures among data points [9].

We propose to estimate the Riemannian metric utilizing similarity measures between data points when data coordinates are available. Suppose that data are generated from an unknown Riemannian manifold, while the Euclidean distance between the coordinates may not reflect the underlying geometry. We model the observed similarity measures between data points as noise-contaminated intrinsic distances, which are then used to characterize the intrinsic geometry on the Riemannian manifold. The targeted Riemannian metric is estimated in a data-driven fashion, which enables path finding via geodesics and calculus on the manifold in which the data reside.

37th Conference on Neural Information Processing Systems (NeurIPS 2023).

This problem is closely related to metric learning [3, 26] where a Mahalanobis distance is commonly used to obtain the best distance for classification [11] and clustering [29] tasks. While a global metric is often the focus of earlier works, multiple local metrics [10, 28, 22, 5] are found to be useful because they better capture the data space geometry. Despite the resemblance, our target is to learn the *Riemannian metric* instead of the *distance metric*, which fundamentally differentiates our proposal from metric learning. We emphasize the data space geometry rather than obtaining a metric optimal for subsequent supervised learning tasks. The additional smoothness of the Riemannian metric tensor field allows analysis of finer structures, while the coefficient matrix for the Mahalanobis distance is locally constant for distance metric learning.

To formulate the problem, let $(\mathcal{M}, G)$ be a Riemannian manifold with Riemannian metric $G$ and induced geodesic distance $dist(\cdot, \cdot)$ which measures the true intrinsic difference between points. Knowing the coordinates of data points $x_0, x_1 \in \mathcal{M}$, we identify each point via its coordinates (a tuple of real numbers). The noisy measures $y$ of the intrinsic distance between data points are referred to as *similarity measures* (equivalently dissimilarity). The response is modeled flexibly, and we consider the following common scenarios: (i) noisy distance, where $y = dist(x_0, x_1)^2 + \epsilon$ for error $\epsilon$, (ii) similarity/dissimilarity, where $y = 0$ if the two points $x_0, x_1$ are considered similar and $y = 1$ otherwise, and (iii) relative comparison, where a triplet of points $(x_0, x_1, x_2)$ are given and $y = 1$ if $x_0$ is more similar to $x_1$ than to $x_2$ and $y = 0$ otherwise. The binary similarity measurement is common in computer vision [e.g. 6], while the relative comparison could be useful for perceptional tasks and recommendation system [e.g. 25, 4]. We aim to estimate the Riemannian metric $G$ and its derivatives using the coordinates and similarity measures among the data points.

The major contribution of this paper is threefold. First, we formulate a framework for probabilistic modeling of similarity measures among data on manifold via intrinsic distances. Such a framework also justifies the distance metric learning from a geometrical perspective and unveils its approximation error for the first time as we know. Second, a theoretical foundation is developed for the proposed method including asymptotic consistency. Last and most importantly, the proposed method provides a geometric interpretation for the structure of the data space induced by the similarity measures, as demonstrated in the numerical examples that include a taxi travel and an MNIST digit application.

## 2 Background in Riemannian Geometry

For brevity, *metric* now refers to *Riemannian metric* while *distance metric* is always spelled out. Throughout the paper, $\mathcal{M}$ denotes a $d$-dimensional manifold endowed with a coordinate chart $(U, \varphi)$, where $\varphi : U \to \mathbb{R}^d$ maps a point $p \in U \subset \mathcal{M}$ on the manifold to its coordinate $\varphi(p) = (\varphi^1(p), \ldots, \varphi^d(p)) \in \mathbb{R}^d$. Without loss of generality, we identify a point by its coordinate as $(p^1, \ldots, p^d)$, suppressing $\varphi$ for the coordinate chart. Upper-script Roman letters denote the components of a coordinate, e.g., $p^i$ is the $i$-th entry in the coordinate of the point $p$, and $\gamma^i$ is the $i$-th component function of a curve $\gamma : \mathbb{R} \supset [a, b] \to \mathcal{M}$ when expressed on chart $U$. The tangent space $T_p\mathcal{M}$ is a vector space consisting of velocities of the form $v = \gamma'(0)$ where $\gamma$ is any curve satisfying $\gamma(0) = p$. The coordinate chart induces a basis on the tangent space $T_p\mathcal{M}$, as $\partial_i|_p = \partial/\partial x^i|_p$ for $i = 1, \ldots, d$, so that a tangent vector $v \in T_p\mathcal{M}$ is represented as $v = \sum_{i=1}^d v^i \partial_i$ for some $v^i \in \mathbb{R}$, suppressing the subscript $p$ in the basis. We adopt the Einstein summation convention unless otherwise specified, namely $v^i \partial_i$ denotes $\sum_{i=1}^d v^i \partial_i$, where common pairs of upper- and lower-indices denotes a summation from 1 to $d$ [see e.g., 18, pp.18–19].

The Riemannian metric $G$ on a $d$-dimensional manifold $\mathcal{M}$ is a smooth tensor field acting on the tangent vectors. At any $p \in \mathcal{M}$, $G(p) : T_p\mathcal{M} \times T_p\mathcal{M} \to \mathbb{R}$ is a symmetric bi-linear tensor/function satisfying $G(p)(v, v) \geq 0$ for any $v \in T_p\mathcal{M}$ and $G(p)(v, v) = 0$ if and only if $v = 0$. On a chart $\varphi$, the metric is represented as a $d$-by-$d$ positive definite matrix that quantifies the distance traveled along infinitesimal changes in the coordinates. With an abuse of notation, the chart representation of $G$ is given by the matrix-valued function $p \mapsto G(p) = [G_{ij}(p)]_{i,j=1}^d \in \mathbb{R}^{d \times d}$ for $p \in \mathcal{M}$, so the distance traveled by $\gamma$ at time $t$ for a duration of $dt$ is $[G_{ij}(\gamma(t))\dot{\gamma}^i(t)\dot{\gamma}^j(t)]^{1/2}$. The intrinsic distance induced by $G$, or the *geodesic distance*, is computed as $dist(p, q) = \inf_\alpha \int_0^1 \sqrt{\sum_{1 \leq i, j \leq d} G_{ij}(\alpha(t))\dot{\alpha}^i(t)\dot{\alpha}^j(t)} dt$ for two points $p, q$ on the manifold $\mathcal{M}$, where infimum is taken over any curve $\alpha : [0, 1] \to \mathcal{M}$ connecting $p$ to $q$.

A *geodesic curve* (or simply *geodesic*) is a smooth curve $\gamma : \mathbb{R} \supset [a, b] \to \mathcal{M}$ satisfying the *geodesic equations*, represented on a coordinate chart as

$$\ddot{\gamma}^k(t) + \dot{\gamma}^i(t)\dot{\gamma}^j(t)\Gamma_{ij}^k \circ \gamma(t) = 0, \text{ for } i, j, k = 1, \ldots, d, \tag{2.1}$$

where over-dots represent derivative w.r.t. $t$; $\Gamma_{ij}^k = \frac{1}{2}G^{kl}\left(\partial_i G_{jl} + \partial_j G_{il} - \partial_l G_{ij}\right)$ are the *Christoffel symbols* at $p$; and $G^{kl}$ is the $(k, l)$-element of $G^{-1}$. Solving (2.1) with initial conditions[1] produces geodesic that traces out the generalization of a straight line on the manifold, preserving travel direction with no acceleration, and is also locally the shortest path.

Considering the shortest path $\gamma$ connecting $p$ to $q$ and applying Taylor's expansion at $t = 0$, we obtain $dist\,(p, q)^2 \approx \sum_{1 \le i, j \le d} G_{ij}(p)(q^i - p^i)(q^j - p^j)$, showing the connection between the geodesic distance the Mahalanobis distance. Our estimation method is inspired by this approximation, and we will discuss the higher-order terms shortly which unveil finer structure of the manifold.

## 3 Local Regression for Similarity Measurements

### 3.1 Probabilistic Modeling for Similarity Measurements

Suppose that we observe $N$ independent triplets $(Y_u, X_{u0}, X_{u1})$, $u = 1, \ldots, N$. Here, the $X_{uj}$ are locations on the manifold identified with their coordinates $\left(X_{uj}^1, \ldots, X_{uj}^d\right) \in \mathbb{R}^d$, $j = 1, 2$, and $Y_u$ are noisy similarity measures of the proximity of $(X_{u0}, X_{u1})$ in terms of the intrinsic geodesic distance $dist\,(\cdot, \cdot)$ on $\mathcal{M}$. To account for different structures of the similarity measures, we model the response in a fashion analogous to generalized linear models. For $X_{u0}, X_{u1}$ lying in a small neighborhood $\mathcal{U}_p \subset \mathcal{M}$ of a target location $p \in \mathcal{M}$, the similarity measure $Y_u$ is modeled as

$$\mathbb{E}\left(Y_u | X_{u0}, X_{u1}\right) = g^{-1}\left(dist\,(X_{u0}, X_{u1})^2\right), \tag{3.1}$$

where $g$ is a given link function that relates the conditional expectation to the squared distance.

*Example* 3.1. We describe below three common scenarios modeled by the general framework (3.1).

1. Continuous response being the squared geodesic distance contaminated with noise:

   $$Y_u = dist\,(X_{u0}, X_{u1})^2 + \sigma(p)\varepsilon_u, \tag{3.2}$$

   where $\varepsilon_1, \ldots, \varepsilon_u$ are i.i.d. mean zero random variables, and $\sigma : \mathcal{M} \to \mathbb{R}^+$ is a smooth positive function determining the magnitude of noise near the target point $p$. This model will be applied to model trip time as noisy measure of cost to travel between locations.

2. Binary (dis)similarity response:

   $$\mathbb{P}\left(Y_u = 1 | X_{u0}, X_{u1}\right) = \text{logit}^{-1}\left(dist\,(X_{u0}, X_{u1})^2 - \hbar\,(p)\right) \tag{3.3}$$

   for some smooth function $\hbar : \mathcal{M} \to \mathbb{R}$, where $\text{logit}(\mu) = \log\left(\mu/\left(1 - \mu\right)\right)$, $\mu \in (0, 1)$ is the logit function. This models the case when there are latent labels for $X_{uj}$ and $Y_u$ is a measure of whether their labels are in common or not. The function $\hbar\,(p)$ in (3.3) describes the homogeneity of the latent labels for points in a small neighborhood of $p$. The latent labels could have intrinsic variation even if measures are made for the same data points $x = X_{u0} = X_{u1}$, and the strength of which is captured by $\hbar\,(p)$.

3. Binary relative comparison response, where we extend our model for triplets of points $(X_{u0}, X_{u1}, X_{u2})$, where $Y_u$ stands for whether $X_{u0}$ is more similar to $X_{u1}$ than to $X_{u2}$:

   $$\mathbb{P}\left(Y_u = 1 | X_{u0}, X_{u1}, X_{u2}\right) = \text{logit}^{-1}\left(dist\,(X_{u0}, X_{u2})^2 - dist\,(X_{u0}, X_{u1})^2\right), \quad \text{(3.4)}$$

   so that the relative comparison $Y_u$ reflects the comparison of squared distances.

---

[1]See Section S3 of the Supplement for details about solving it in practice.

## 3.2 Local Approximation of Squared Distances

It is the squared distance that determines the responses in our model (3.1), which is justified by the following local approximation. Proposition 3.1 provides a Taylor expansion for the squared geodesic distance between two geodesics with same starting point but different initial velocities (see Figure S5.1 for visualization). It is the key tool to estimate model (3.1) through local regression. Furthermore, (3.5) characterize the error of approximating geodesic distance with Mahalanobis distance. For a point $p$ on the Riemannian manifold $\mathcal{M}$, let $\exp_p : T_p\mathcal{M} \rightarrow \mathcal{M}$ denote the exponential map defined by $\exp_p(tv) = \gamma(t)$ where $\gamma$ is a geodesic starting from $p$ at time 0 with initial velocity $\gamma'(0) = v \in T_p\mathcal{M}$. For notational simplicity, we suppress the dependency on $p$ in geometric quantities (e.g., the metric $G$ is understood to be evaluated at $p$). For $i = 1, \ldots, d$, denote $\delta^i = \delta^i(t) = \gamma^i(t) - \gamma^i(0)$ as the difference in coordinate after a travel of time $t$ along $\gamma$.

**Proposition 3.1** (spread of geodesics, coordinated)**.** Let $p \in \mathcal{M}$ and $v, w \in T_p\mathcal{M}$ be two tangent vectors at $p$. On a local coordinate chart, the squared geodesic distance between two geodesics $\gamma_0(t) = \exp_p(tv)$ and $\gamma_1(t) = \exp_p(tw)$ satisfies, as $t \rightarrow 0$,

$$dist\left(\gamma_0(t), \gamma_1(t)\right)^2 = \delta_{0-1}^i \delta_{0-1}^j G_{ij} + \delta_{0-1}^i \left(\delta_0^k \delta_0^l - \delta_1^k \delta_1^l\right) \Gamma_{kl}^j G_{ij} + O(t^4), \quad (3.5)$$

where for $i, j, k, l, m = 1, \ldots, d$,

- $\delta_0^i = \gamma_0^i(t) - p^i$, $\delta_1^i = \gamma_1^i(t) - p^i$, and $\delta_{0-1}^i = \delta_0^i - \delta_1^i$, i.e., $\delta_0^i, \delta_1^i$ are differences in $i$-th coordinates of $\gamma_0(t)$ and $\gamma_1(t)$ to the origin $p$, respectively, and $\delta_{0-1}^i = \delta_0^i - \delta_1^i$ is the coordinate difference between $\gamma_0(t)$ and $\gamma_1(t)$;

- $G_{ij}$ and $\Gamma_{kl}^j$ are the elements of the metric and Christoffel symbols at $p$, respectively.

*Proof.* See Section S5 in the Supplement. $\square$

To the RHS of (3.5), the first term is the quadratic term in distance metric learning. The second term is the result of coordinate representation of geodesics. It vanishes under the *normal coordinate* where the Christoffel symbols are zero.[2] It inspires the use of local regression to estimate the metric tensor and the Christoffel symbols. For $X_{u0}, X_{u1}$ in a small neighborhood of $p$, write the linear predictor as

$$\eta_u := \beta^{(0)} + \delta_{u,0-1}^i \delta_{u,0-1}^j \beta_{ij}^{(1)} + \delta_{u,0-1}^k \left(\delta_{u0}^i \delta_{u0}^j - \delta_{u1}^i \delta_{u1}^j\right) \beta_{ijk}^{(2)}, \quad (3.6)$$

a function of the intercept $\beta^{(0)}$ and coefficients $\beta_{ij}^{(1)}, \beta_{ijk}^{(2)}$, where $\delta_{u0}^i = X_{u0}^i - p^i$, $\delta_{u1}^i = X_{u1}^i - p^i$, and $\delta_{u,0-1}^i = \delta_{u0}^i - \delta_{u1}^i$, for $i, j, k, l = 1, \ldots, d$, and $u = 1, \ldots, N$. The intercept term $\beta^{(0)}$ is included for capturing intrinsic variation (e.g., $\hbar(p)$ in (3.3)) and can otherwise be dropped. The link function connects the linear predictor to the conditional mean via $\mu_u := g^{-1}(\eta_u) \approx \mathbb{E}(Y_u | X_{u0}, X_{u1})$ as indicated by (3.1) and (3.5), where $\mu_u$ is seen as a function of the coefficients $\beta^{(0)}$, $\beta_{ij}^{(1)}$, and $\beta_{ijk}^{(2)}$. Therefore, upon the specification of a loss function $Q : \mathbb{R} \times \mathbb{R} \rightarrow \{0\} \cup \mathbb{R}^+$ and non-negative weights $w_1, \ldots, w_N$, the minimizers

$$(\hat{\beta}^{(0)}, \hat{\beta}_{ij}^{(1)}, \hat{\beta}_{ijk}^{(2)}) = \underset{\beta^{(0)}, \beta_{ij}^{(1)}, \beta_{ijk}^{(2)}; i,j,k}{\arg\min} \sum_{u=1}^{N} Q(Y_u, \mu_u) w_u, \quad (3.7)$$

subject to

$$\beta_{ij}^{(1)} = \beta_{ji}^{(1)}, \ \beta_{ijk}^{(2)} = \beta_{jik}^{(2)}, \text{ for } i, j, k, l = 1, \ldots, d, \quad (3.8)$$

are used to estimate the metric tensor and Christoffel symbols, obtaining

$$\hat{G}_{ij} = \hat{\beta}_{ij}^{(1)}, \quad \hat{\Gamma}_{ij}^l = \hat{\beta}_{ijk}^{(2)} \hat{G}^{kl}, \quad (3.9)$$

where $\hat{G}^{kl}$ is the matrix inverse of $\hat{G}$ satisfying $\hat{G}^{kl}\hat{G}_{kj} = 1_{\{j=l\}}$. The symmetry constraints (3.8) are the result of the symmetries in the metric tensor and Christoffel symbols, and are enforced by optimizing over only the lower triangular indices $1 \leq i < j \leq d$ without constraints. Asymptotic

---

[2]See e.g., [19] pp. 131–133 for normal coordinate. See [20] and Proposition S5.1 of the Supplement for coordinate-invariant version of Proposition 3.1.

results provide the positive-definiteness of the metric estimate, as will be shown in Proposition 4.1. To weigh the pairs of endpoints according to their proximity to the target location $p$, we apply kernel weights specified by

$$w_u = h^{-2d} \prod_{i=1}^{d} K\left(\frac{X_{u0}^i - p^i}{h}\right) K\left(\frac{X_{u1}^i - p^i}{h}\right) \tag{3.10}$$

for some $h > 0$ and non-negative kernel function $K$. The bandwidth $h$ controls the bias–variance trade-off of the estimated Riemannian metric tensor and its derivatives.

Altering the link function $g$ and the loss function $Q$ in (3.7) enables flexible local regression estimation for models in Example 3.1.

*Example* 3.2. Consider the following loss functions for estimating the metric tensors and the Christoffel symbols when data are drawn from model (3.2)–(3.4), respectively.

1. Continuous noisy response: use squared loss $Q(y, \mu) = (y - \mu)^2$ with $g$ being the identity link function so $\mu_u = \eta_u$.

2. Binary (dis)similarity response: use log-likelihood of Bernoulli random variable

$$Q(y, \mu) = y \log \mu + (1 - y) \log (1 - \mu), \tag{3.11}$$

and $g$ the logit link, so $\mu_u = \text{logit}^{-1}(\eta_u)$. The model becomes a local logistic regression.

3. Binary relative comparison response: apply the same loss function (3.11) and logit link as in the previous scenario, but here we formulate the linear predictor based on $dist(X_{u0}, X_{u2})^2 - dist(X_{u0}, X_{u1})^2 \approx \eta_{u1} - \eta_{u2}$ and

$$\mu_u = g^{-1}(\eta_{u1} - \eta_{u2}). \tag{3.12}$$

Locally, the difference in squared distances is approximated by

$$\eta_{u1} - \eta_{u2} = \left(\delta_{u,0-1}^i \delta_{u,0-1}^j - \delta_{u,0-2}^i \delta_{u,0-2}^j\right) \beta_{ij}^{(1)} + \left(\delta_{u,0-1}^k \left(\delta_{u0}^i \delta_{u0}^j - \delta_{u1}^i \delta_{u1}^j\right)\right) \beta_{ijk}^{(2)}$$
$$- \left(\delta_{u,0-2}^k \left(\delta_{u0}^i \delta_{u0}^j - \delta_{u2}^i \delta_{u2}^j\right)\right) \beta_{ijk}^{(2)}, \tag{3.13}$$

for $\delta_{u2}^i = X_{u2}^i - p^i$ and $\delta_{u,0-2}^i = \delta_{u2}^i - \delta_{u0}^i$, $i = 1, \ldots, d$. Here $\eta_{u1}$ and $\eta_{u2}$ are constructed in analogy to (3.6) using $(X_{u0}, X_{u1})$ and $(X_{u0}, X_{u2})$ pair respectively.

Examples in Section 5 will further illustrate the proposed method in those scenarios. Besides the models listed, other choices for the link $g$ and loss function $Q$ can also be considered under this local regression framework [8], accommodating a wide variety of data. To efficiently estimate the metric on the entire manifold $\mathcal{M}$, we apply a procedure based on discretization and post-smoothing, as detailed in Section S3 of the Supplement. In short, kernel smoothing (weighted average) of estimated component functions $\hat{G}_{ij}$ over some grid points provides a smooth tensor field, which ease the burden of re-estimation for every points on the manifold.

## 4  Bias and Variance of the Estimated Metric Tensor

This subsection provides asymptotic justification for model (3.2) with $\mathbb{E}(Y_u | X_{u0}, X_{u1}) = dist(X_{u0}, X_{u1})^2$ under the squared loss $Q(\mu, y) = (\mu - y)^2$ and the identity link $g(\mu) = \mu$. The estimator we analyzed here fits a local quadratic regression without intercept and approximates the squared distance by a simplified form of (3.6):

$$dist(X_{u0}, X_{u1})^2 \approx \eta_u := \delta_{u,0-1}^i \delta_{u,0-1}^j \beta_{ij}^{(1)}, \tag{4.1}$$

for $u = 1, \ldots, N$. Given a suitable order of the indices $i, j$ for vectorization, we rewrite the formulation into a matrix form. Denote the local design matrix and regression coefficients as

$$\mathbf{D}_u = \left(\delta_{u,0-1}^1 \delta_{u,0-1}^1, \ldots, \delta_{u,0-1}^i \delta_{u,0-1}^j, \ldots, \delta_{u,0-1}^d \delta_{u,0-1}^d\right)^T,$$
$$\boldsymbol{\beta} = \left(\beta_{11}^{(1)}, \ldots, \beta_{ij}^{(1)}, \ldots, \beta_{dd}^{(1)}\right)^T,$$

so that the linear predictor $\eta_u = \mathbf{D}_u^T \boldsymbol{\beta}$. Further, write $\mathbf{D} = (\mathbf{D}_1, \ldots, \mathbf{D}_N)^T$, $\mathbf{Y} = (Y_1, \ldots, Y_N)^T$, and $\mathbf{W} = \mathrm{diag}\,(w_1, \ldots, w_N)$, with weights $w_u$ specified in (3.10). The objective function in (3.7) becomes $(\mathbf{Y} - \mathbf{D}\boldsymbol{\beta})^T \mathbf{W} (\mathbf{Y} - \mathbf{D}\boldsymbol{\beta})$, and the minimizer is $\hat{\boldsymbol{\beta}} = (\mathbf{D}^T \mathbf{W} \mathbf{D})^{-1} \mathbf{D}^T \mathbf{W} \mathbf{Y}$, for which we will analyze the bias and variance.

To characterize the asymptotic bias and variance of the estimator, we assume the following conditions are satisfied in a neighborhood of the target $p$. These conditions are standard and analogous to those assumed in a local regression setting [8].

(A1) The joint density of endpoints $(X_{u0}, X_{u1})$ is positive and continuously differentiable.

(A2) The functions $G_{ij}, \Gamma_{ij}^k$ are $C^2$-smooth for $i, j, k = 1, \ldots, d$.

(A3) The kernel $K$ in weights (3.10) is symmetric, continuous, and has a bounded support.

(A4) $\sup_u \mathrm{var}\,(Y_u | X_{u0}, X_{u1}) < \infty$.

**Proposition 4.1.** Under (A1)–(A4), $\mathrm{bias}\left(\hat{\boldsymbol{\beta}} | \mathbf{X}\right) = O_p\left(h^2\right)$, $\mathrm{var}\left(\hat{\boldsymbol{\beta}} | \mathbf{X}\right) = O_p\left(N^{-1} h^{-4-2d}\right)$, as $h \to 0$ and $N h^{2+2d} \to \infty$, where $\mathbf{X} = \{(X_{u0}, X_{u1})\}_{u=1}^N$ is the collection of observed endpoints.

*Proof.* See Section S6 in the Supplement. $\qquad\square$

The local approximation (4.1) is similar to a local polynomial estimation of the second derivative of a $2d$-variate squared geodesic distance function, explaining the order of $h$ in the rates for bias and variance.

The positive definite constraints (as in distance metric learning) are dropped based on the asymptotic properties, thus the estimated metric could fail to be positive definite at some locations. This is alleviated by sufficient data and post-smoothing that averaging neighboring estimation.

## 5 Simulation

We illustrate the proposed method using simulated data with different types of responses as described in Example 3.1. We study whether the proposed method well estimates Riemannian geometric quantities, including the metric tensor, geodesics, and Christoffel symbols. Additional details including assessment on stability and positive-definiteness are included in Section S4 of the Supplement. Subsection S4.2 also provides an ellipsoid example for manifold of non-constant curvature.

### 5.1 Unit Sphere

The usual arc-length/great circle distance on the $d$-dimensional unit sphere is induced by the round metric, which is expressed under the stereographic projection coordinate $(x^1, \ldots, x^d)$ as $\mathring{G}_{ij} = 4\left(1 + \sum_{k=1}^d x^k x^k\right)^{-2} 1_{\{i=j\}}$, for $i, j = 1, \ldots, d$. Under the additive model (3.2) in Example 3.1, we considered either noiseless or noisy responses by setting $\sigma(p) = 0$ or $\sigma(p) > 0$ respectively.

Experiments were preformed with $d = 2$ and the finding are summarized in Figure 5.1. For continuous responses, the left panel of Figure 5.1a visualizes the true and estimated metric tensors via cost ellipses (S2.1) and the right panel shows the corresponding geodesics by solving the geodesic equations (2.1) with true and estimated Christoffel symbols. The metrics and the geodesics were well estimated under the continuous response model with or without additive errors, where the estimates overlap with the truth. Figure 5.1c evaluates the relative estimation errors $\left\|\hat{G} - G\right\|_F / \|G\|_F$ and $\left\|\hat{\Gamma} - \Gamma\right\|_F / \|\Gamma\|_F$ w.r.t. the Frobenius norm (S2.2) for data from the continuous model (3.2).

For binary responses under model (3.3), Figure 5.1b visualizes the data where the background color illustrates $\hbar$. Figure 5.1d and left panel of Figure 5.1a suggest that the intercept and the metric were reasonably estimated, while the geodesics are slightly away from the truth (Figure 5.1a, right). This indicates that the binary model has higher complexity and less information is provided by the binary response (see also Figure S4.1b in the Supplement).

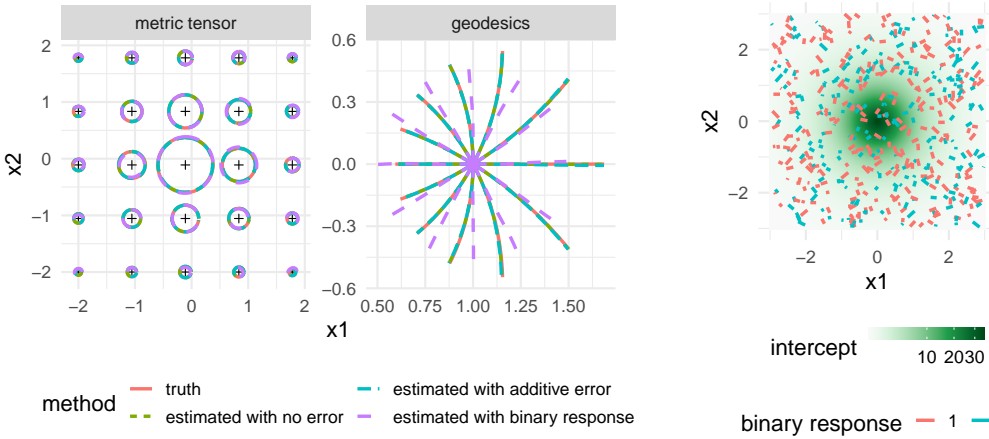

(a) Estimated and true metric tensors using ellipses representation (left) and the geodesic curves (right) starting from $(1,0)$ with unit initial velocities pointing to 1–12 o'clock directions.

(b) Binary responses, with segments showing comparisons, background colored to $\hbar$.

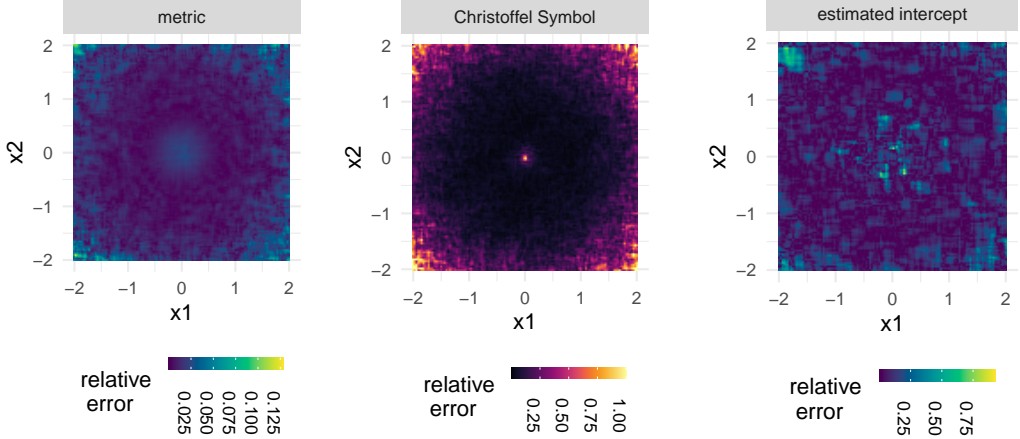

(c) Relative errors in term of Frobenius norm of the estimated tensors for the continuous response model (3.2) with additive error.

(d) Errors for estimating $\hbar$ with binary responses (3.3).

Figure 5.1: Simulation results for 2-dimensional sphere under stereographic projection coordinate.

## 5.2 Relative Comparison on the Double Spirals

A set of $7 \times 10^4$ points on $\mathbb{R}^2$ were generated around two spirals, corresponding to two latent classes $\mathcal{A}$ and $\mathcal{B}$ (e.g., green points in Figure 5.2a are from latent class $\mathcal{A}$). We compare neighboring points $(X_{u0}, X_{u1}, X_{u2})$ to generate relative comparison response $Y_u$ as follows. For $u = 1, \ldots, N$, $Y_u = 1$ if $X_{u0}, X_{u1}$ belong to the same latent class and $X_{u0}, X_{u2}$ belong to different classes; otherwise $Y = 0$. Figure 5.2b shows a portion of the $N = 6,965,312$ comparison generated, where the hollow circles in the middle of each wedge correspond to $X_{u0}$.

Here, contrast of the two latent classes induces the intrinsic distance, so the distance is larger across the supports of the two classes and smaller within a single support. Therefore, the resulting metric tensor should reflect less cost while moving along the tangential direction of the spirals compared to perpendicular directions. Estimates were drawn under model (3.4) by minimizing the objective (3.7) with the link function (3.12) and the local approximation (3.13).

The estimated metric shown in Figure 5.2c is consistent with the interpretation of the intrinsic distance and metric induced by the class membership discussed above. Meanwhile, the estimated geodesic curve unveils the hidden circular structure of the data support as shown in Figure 5.2d.

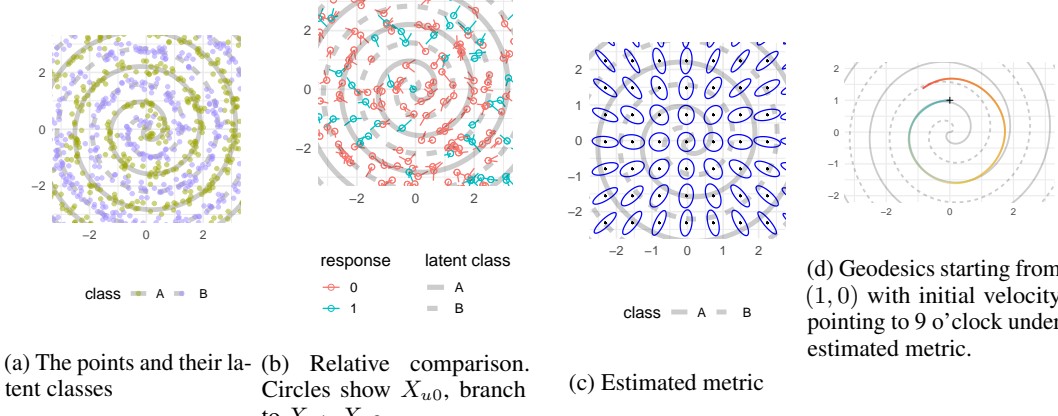

(a) The points and their latent classes

(b) Relative comparison. Circles show $X_{u0}$, branch to $X_{u1}, X_{u2}$.

(c) Estimated metric

(d) Geodesics starting from $(1, 0)$ with initial velocity pointing to 9 o'clock under estimated metric.

Figure 5.2: Relative comparison on double spirals. Gray curves (solid and dashed) in the background represent the approximate support of the two latent classes.

## 6  New York City Taxi Trip Duration

We study the geometry induced by taxi travel time in New York City (NYC) during weekday morning rush hours. New York City Taxi and Limousine Commission (TLC) Trip Record Data was accessed on May 1st, 2022[3] to obtained business day morning taxi trip records including GPS coordinates for pickup/dropoff locations as $(X_{u0}, X_{u1})$ and trip duration as $Y_u$. Estimation to the travel time metric was drawn under model (3.2) with $Q(y, \mu) = (y - \mu)^2$ and $g(\mu) = \mu$.

Figure 6.1a shows the estimated metric for taxi travel time. The background color shows the Frobenius norm of the metric tensor, where larger values mean that longer travel time is required to pass through that location. Trips through midtown Manhattan and the financial district were estimated to be the most costly during rush hours, which is coherent to the fact that these are the city's primary business districts. Moreover, the cost ellipses represent the cost in time to travel a unit distance along different directions. This suggests that in Manhattan, it takes longer to drive along the east–west direction (narrower streets) compared to the north–south direction (wider avenues).

Geodesic curves in Figure 6.1b show where a 15-minutes taxi ride leads to starting from the Empire State Building. Each geodesic curve corresponds to one of 12 starting directions (1–12 o'clock). Note that we apply a continuous Riemannian manifold approximation to the city, so the geodesic curves provide approximations to the shortest paths between locations and need not conform to the road network. Travel appears to be faster in lower Manhattan than in midtown Manhattan. The spread of the geodesics differs along different directions, indicating the existence of non-constant curvature on the manifold and advocating for estimating the Riemannian metric tensor field instead of applying a single global distance metric.

## 7  High-dimensional Data: An Example with MNIST

The curse of dimensionality is challenging for nonparametric methods applied on data sources like images and audios. However, it is often found that apparent high-dimensional data actually lie close to some low-dimensional manifold, which is utilized by manifold learning literature to produce reasonable low-dimensional coordinate representations. The proposed method can be applied to the resulting low-dimensional coordinates to recover the high-dimensional geometry even implicitly defined, demonstrated in the following MNIST example.

We embed images in MNIST to a 2-dimensional space via tSNE [13]. Similarity between the objects was computed by the sum of the Wasserstein distance between images[4] and the indicator of whether the underlying digits are different (1) or not (0). The goal is to infer the geometry of the embedded

---

[3]https://www1.nyc.gov/site/tlc/about/tlc-trip-record-data.page. Data format changed after our download.

[4]After rescaling, see Subsection S4.5 of the Supplement.

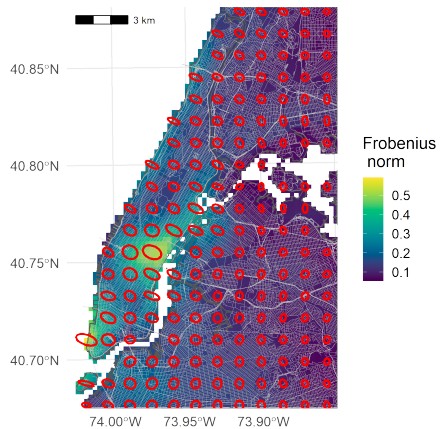

(a) Estimated metric tensors for trip duration: cost ellipses and Frobenius norm (background color).

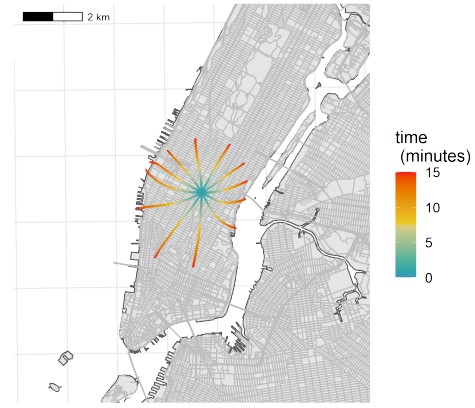

(b) Geodesics correspond to 15-minute taxi rides from the Empire State Building heading to 1–12 o'clock.

Figure 6.1: New York taxi travel time during morning rush hours.

data induced by this similarity measures. Estimation was drawn under model (3.2) with squared loss $Q(y, \mu) = (y - \mu)^2$ and the identity link. An intercept term ($\beta^{(0)}$ in (3.6)) is used to capture intrinsic variation partially due to the non-injective dimensional reduction.

The geodesics estimated from our method tend to minimize the number of switches between labels. For example, the geodesic A in Figure 7.1 remains "4" (1st row of panel (b)) throughout, while the straight line on the tSNE chart translates to a path of images switching between "4" and "9" (2nd row of panel (b)); similar phenomenon occurs for geodesics B and C (corresponding to 7th and 12th rows in (b)). Also, our estimated geodesics produce reasonable transition and reside in the space of digits, while unrestricted optimal transport (3rd, 8th, and 13th rows of panel (b)) could produce unrecognizable intermediate images. Our estimated geodesic is faithful to the geometric interpretation that a geodesic is locally the shortest path. Moreover, our proposal provide sensible image transition, especially compared to the shortest paths along neighborhood graph in the embedded space (dotted blue lines in (a) and 4, 9, 14 rows in (b)) or kNN graphs in the original Euclidean space $\mathbb{R}^{28 \times 28}$ (5, 10, 15 rows in (b)). This could be useful for complex data interpolation that preserves geometry.

## 8 Discussion

We present a novel framework for inferring the data geometry based on pairwise similarity measures. Our framework targets data lying on a low-dimensional manifold where observations are expected to be dense near the locations where we wish to estimate the metric. However, these assumptions are general enough for our method to be applied to manifold data with high ambient dimension in combination with manifold embedding tools. Context-specific interpretation of the geometrical notions, e.g., for Riemannian metric and geodesics, has been demonstrated in the taxi travel and MNIST digit examples. Our method could inspire additional compelling applications to other topics such as cognition and perception research, where psychometric similarity measures are commonly made. Moreover, further development including (but not limited to) handling sparse data and ensuring positivity in the Riemannian metric estimation is needed, as discussed in Section S1 of the Supplement.

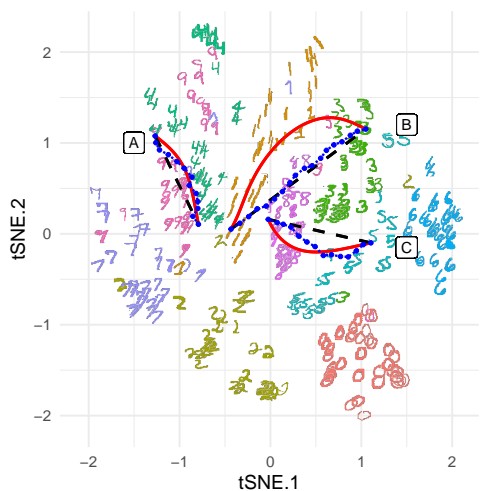

(a) The estimated geodesic curves (solid red), shortest path along graph (dotted blue), and straight lines on the chart (dashed black).

(b) Image transitions corresponding to A, B, and C in (a). Every 5 rows correspond to a set of paths sharing the same pair of starting and ending images, where the 1st – 5th row correspond to the estimated geodesics, the straight lines on the chart, the optimal transport (path not shown in (a)), the paths along neighborhood graph on the chart, and along kNN graph in the Euclidean space (path not shown in (a)), respectively.

Figure 7.1: Geometry induced by a sum of Wasserstein distance and same-digit-or-not indicator.

## Acknowledgments and Disclosure of Funding

The authors would like to thank Dr. Zhengyuan Zhu and the reviewers for the helpful feedback and discussion. The research was partially supported by NSF grant DMS-2329879.

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
