# Supplement to "Estimating Riemannian Metric with Noise-Contaminated Intrinsic Distance"

Jiaming Qiu[1] and Xiongtao Dai[2]

[1]Fred Hutchinson Cancer Center
[2]University of California, Berkeley

## Contents

# S1 Limitations and Potential Further Development

Unlike distance metric learning where the subsequent tasks utilizing the estimated distance metric is the usual focus, the proposal focuses on the estimated metric characterizing the geometry structure. Despite the illustrated taxi and MNIST examples, it is still open to finding more compelling applications that target the data space geometry. Interpreting mathematical concepts such as Riemannian metric and geodesic in the context of potential application (e.g., cognition and perception research where similarity measures are common) could be inspiring.

Our proposal requires sufficiently dense data, which could be demanding, especially for high-dimensional data due to the curse of dimensionality. Dimensional reduction (e.g., manifold embedding as in the MNIST example) can substantially alleviate the curse of dimensionality, and the dense data requirement will more likely hold true.

The proposed estimator does not guarantee a positive definite metric. The positive-definiteness of the estimator is justified by the large data size and the asymptotic consistency (Proposition 4.1). We inspect the positive-definiteness of the estimated metrics in Subsection S4.6, where most are with a few exceptions near the boundary due to the lack of neighboring observations.

Moreover, the estimated Christoffel symbol is not guaranteed to be the derivative of the estimated metric, which is common in local polynomial derivatives estimation. Theoretically, the estimated derivative should reflect the true derivative rather than being the derivative of the estimated curve. In practice, this may be annoying, especially for geodesic computation where numeric error accumulates fast while solving ODE. In such a situation, we recommend imposing further post-smoothing, or utilizing the numeric derivative of the estimated metric (as in the MNIST example, see Subsection S4.5). We consider this an issue open for further development beyond the scope of this paper.

We also encourage the readers to refer to `https://openreview.net/forum?id=VhLU3pStsl` for many helpful reviews and discussion. We are deeply grateful for the reviewers' time and consideration.

# S2 Additional Definition

A *cost ellipse* visualizes the metric by an ellipse

$$\mathscr{E}_p = \left\{ \left(x^1, \ldots, x^d\right) : \sum_{i,j=1}^{d} \left(x^i - p^i\right)\left(x^j - p^j\right) G^{ij} = r^2 \right\} \tag{S2.1}$$

for some constant $r > 0$, which shows, approximately, the intrinsic distance on the manifold when traveling a unit length on the coordinate chart along each direction. More precisely, it shows the norm of tangent vectors $v^i \partial_i \in T_p \mathcal{M}$

subject to $\sum_{i=1}^{d} v^i v^i = r^2$ at $p$ pointing to the corresponding direction. For example in a $d = 2$-dimensional manifold at $p = 0$, under $G = \text{diag}(\lambda_1, \lambda_2)$ with $\lambda_1 > \lambda_2$ and $r = 1$. The long axis, $\left(\pm\sqrt{\lambda_1}, 0\right)$, is the norm of the tangent vector $\pm\partial_1$. Thus, the direction in which the ellipse is larger corresponds to the direction of the larger geodesic distance. One can see (S2.1) as the "inverse" of the Tissot's indicatrix, where the latter shows a local equidistance contour to the ellipse's center.

*Frobenius norm* of tensors is denoted as $\|\cdot\|_F$, defined as

$$\|G\|_F = \left( \sum_{i,j=1}^{d} G_{ij} G_{ij} \right)^{1/2}, \quad \|\Gamma\|_F = \left( \sum_{i,j,k=1}^{d} \Gamma_{ij}^k \Gamma_{ij}^k \right)^{1/2}, \qquad \text{(S2.2)}$$

for metric tensor $G$ and Christoffel symbol $\Gamma$.

## S3 Implementation Notes

An R (R Core Team, 2022) package is developed to implement the proposed methods and all numerical experiments. It is available at https://github.com/jiamingqiu/remeloc. A desktop computer (12-core 24-thread CPU, 128 GB RAM) is used to run all experiments. Most experiments were finished within several hours.

We utilized an efficient procedure to obtain estimates $\hat{G}_{ij}$ over the entire manifold $\mathcal{M}$ as follows. We first obtain estimates $\hat{G}_{ij}(p_n)$ over a dense grid of points $p_1, p_2, \ldots, p_{\texttt{n\_grid}} \in \mathcal{M}$ by following (3.7)–(3.9). Next, the estimate $\hat{G}_{ij}(x)$ at an arbitrary $x \in \mathcal{M}$ is obtained by the post-smoothing estimate

$$\hat{G}_{ij}(x) = \frac{\sum_{n=1}^{\texttt{n\_grid}} K\left(\|x - p_n\| / h_{\text{ps}}\right) \hat{G}_{ij}(p_n)}{\sum_{n=1}^{\texttt{n\_grid}} K\left(\|x - p_n\| / h_{\text{ps}}\right)},$$

for some kernel $K$ and bandwidth $h_{\text{ps}} > 0$. Note that it is the component functions $\hat{G}_{ij}$ that we average, but not the metric tensor $G_{ij} dx^i dx^j$. The post-smoothing process averages neighboring estimates to create a smoother tensor field, and also alleviate potential positive-definiteness issues. Local regression (Loader, 1999) for post-smoothing could also be used. The grid for the examples (Section 5, Section 6, and Section 7) are $128 \times 128$ for the unit sphere, and $80 \times 80$ for the double spirals, $250 \times 250$ meters cells for the New York taxi example, and $64 \times 64$ for the MNIST example.

The estimated geodesics are computed by numerically solving ordinary differential equations system, either given the start point and initial velocity, or given the start and the end points. It suffices to notice that the geodesic equations (2.1) are equivalently written as, after plugging-in the estimated Christoffel symbol $\hat{\Gamma}$,

$$v^i(t) = \dot{\gamma}^i(t),$$
$$\dot{v}^k(t) = -v^i(t) v^j(t) \hat{\Gamma}_{ij}^k \circ \gamma(t),$$

for $i, j, k = 1, \ldots, d$. Here, $\hat{\Gamma}_{ij}^k \circ \gamma(t)$ is the value of the estimated Christoffel symbol at point $(\gamma^1(t), \ldots, \gamma^d(t))$. Further supplying initial condition $\gamma^i(0) = p_0^i$, $v^i(0) = v_0^i, i = 1, \ldots, d$ for point $p_0 \in \mathcal{M}$ and tangent vector $v_0 \in T_{p_0}\mathcal{M}$ constitute an initial value problem, whose solution reflects the geodesic curve starting from $p_0$ with initial velocity $v_0$. On the other hand, supplying boundary condition $\gamma^i(0) = p_0^i$, $\gamma^i(1) = p_1^i, i = 1, \ldots, d$ for $p_0, p_1 \in \mathcal{M}$ constitute a boundary value problem, whose solution reflects the geodesic curve from $p_0$ to $p_1$. we use `deSolve` (Soetaert et al., 2010) and `bvpSolve` (Mazzia et al., 2014) for initial value problems and boundary value problems respectively. See reference therein for further details of numeric solution to ODE.

# S4 Additional Experiment Details

This section provides further detail to complete Section 5, Section 6, and Section 7 of the main text including how we generated the simulated data, and more figures.

## S4.1 Unit Sphere and Round Metric

The stereographic coordinate of the $d$-dimensional sphere $\mathbb{S}^d$ identifies points on the sphere by mapping it to its stereographic projection in $\mathbb{R}^d$ from the north pole. The round metric on the sphere $\mathbb{S}^d$ is the metric induced by embedding of $\mathbb{S}^d \hookrightarrow \mathbb{R}^{d+1}$. For detail, see for example page 30 of Lee (2013) and chapter 3 of Lee (2018). We generated endpoints $X_{u0}, X_{u1}$ uniformly in the coordinate chart $(-3, 3) \times (-3, 3)$, then pair the endpoints so that the difference in coordinates of the endpoints $|\delta_{u0}^i - \delta_{u1}^i| = |X_{u0}^i - X_{u1}^i|$ would not exceed 0.2 for $i = 1, 2$, $u = 1, \ldots, N$.

More precisely, data were generated on $d = 2$-dimensional sphere under stereographic projection coordinate. A total of $N = 5 \times 10^5$ pairs of endpoints with $X_{u0}^i, X_{u1}^i \in (-3, 3)$ were generated subject to $|X_{u0}^i - X_{u1}^i| \leq 0.2$ for all $u = 1, \ldots, N; i = 1, \ldots, d$. For a reasonable signal-to-noise ratio, we set $\sigma(p) = \sigma \approx 9 \times 10^{-4}$ for all $p$, which is approximately one-tenth of the marginal expectation of squared distance, i.e., $\sigma \approx \mathbb{E} \, dist \, (X_{u0}, X_{u1})^2 / 10$.

For simplicity of presentation, we scaled the distance for the binary similarity response model (3.3). More precisely, we use $dist_c (\cdot, \cdot) = \sqrt{c} \, dist (\cdot, \cdot)$ induced by the scaled metric $G_{ij,c} = c \mathring{G}_{ij}$ for some constant $c$ and $i, j = 1, \ldots, d$. The experiment here used $c = 300$. Intuitively, the constant $c$ regulates the signal-to-noise ratio without changing the form of geodesics. Given the endpoints, a smaller $c$ leads to a smaller value of geodesic distance and hence smaller variation in the linear predictors $\eta_u$, so the response $Y_u$ will take less influence from the distance, representing a higher amount of noise. Then $\hbar(p)$ was set to be the average local squared distances within a local neighborhood of $p$ under the scaled distance.

In the end, the responses were generated following (3.2) and (3.3) respectively.

In addition, Figure S4.1 illustrates the relative Frobenius error for estimated tensors using noiseless or binary responses.

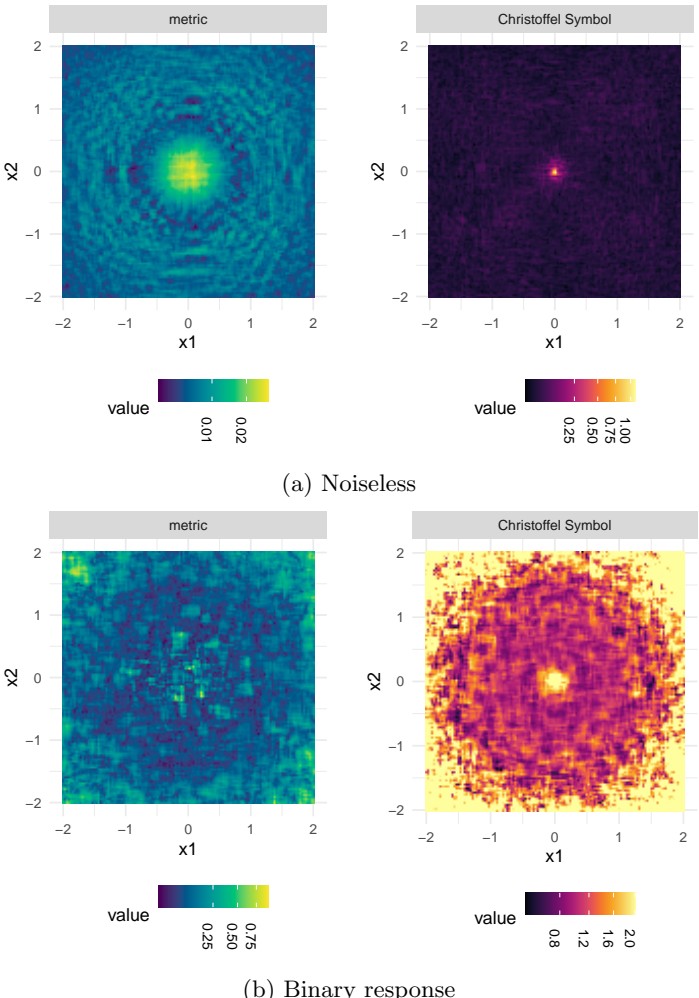

(a) Noiseless

(b) Binary response

Figure S4.1: Relative errors w.r.t. Frobenius norm (S2.2) of the estimated tensors with noiseless or binary response for 2-dimensional sphere under stereographic projection coordinate chart.

### S4.1.1 Bandwidth Selection

Like local regression, the proposed method relies on a neighborhood specification for optimal bias-variance trade-off. The simulation in Subsection 5.1 uses the rectangular kernel $K(x) = \mathbf{1}_{[-1,1]}(x)$ for (3.10), where $\mathbf{1}$ is the indicator function, so the estimation only utilizes observations with endpoints $X_{u0}, X_{u1}$ are both

lying in the neighborhood $\mathcal{U}_p = \left\{ \left(x^1, \ldots, x^d\right) : |x^i - p^i| \leq h, i = 1, \ldots, d \right\}$ of the target point $p$.

We propose a train–test set scheme for data-driven bandwidth selection. To simplify computation, we only considered additive error under (3.2). A $16 \times 16$ grid $p_1, \ldots, p_{256} \in (-3, 3) \times (-3, 3)$ were used as target points where metric tensors were estimated, with a test set of $N_{\text{test}} = 31246$ observations that were within close proximity to the grid. Estimation of the tensors were computed w.r.t. bandwidth $h$ utilizing a train set containing $N_{\text{train}} = 400158$ (approximately 80% of the data) randomly selected observations outside of the test set. For the test set, $\hat{Y}_{u,\text{test}} = \hat{\eta}_{u,\text{test}}$ were then computed under identity link by plugging the estimated tensors into (3.6).

The bandwidth minimizing the squared loss $\sum_{u=1}^{N_{\text{test}}} \left(Y_{u,\text{test}} - \hat{Y}_{u,\text{test}}\right)^2 / N_{\text{test}}$ is then chosen. The proposed bandwidth selection resulted in $h = 0.18$ according to the loss shown in the left panel of Figure S4.2, which corresponds to an effectively local sample size of approximately 1000 in the training step (right panel of Figure S4.2). These tuning parameter choices were applied in the results shown in Subsection 5.1. Other bandwidth selection methods developed for local regression (see e.g. Fan and Gijbels, 1996, section 4.10) can also be adopted here.

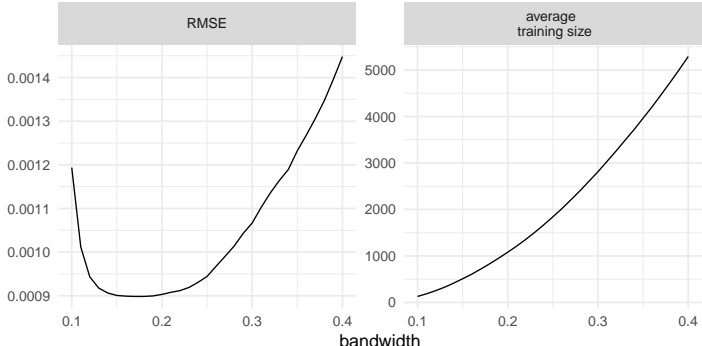

Figure S4.2: Root mean squared error (RMSE) for the test set (Left) and the average number of local training observations (Right) as the bandwidth $h$ varies.

### S4.1.2   Error Bar

The performance is stable among data generated following the same setup w.r.t. different random seeds. Figure S4.3 shows the fluctuation of Frobenius errors over 15 repeated experiments. Let $\hat{G}_{(i)}(p)$ denotes the estimated metric tensor at point $p$ during the $i$-th repeat. Compute the sample standard deviation $\sigma(p)$ and mean $\mu(p)$ of $\left\| \hat{G}_{(1)}(p) - G(p) \right\|_F, \ldots, \left\| \hat{G}_{(N^*)}(p) - G(p) \right\|_F$, where $G$ is the true round metric, $N^*$ is the number of repeated experiments. Then the ratio $\sigma(p)/\mu(p)$ serves as a relative quantification of the variation of estimation

errors. Figure S4.3 shows that the proposal is stable, and so are the reported errors in the main text. Note that there are more variation in the errors when observations are noisier.

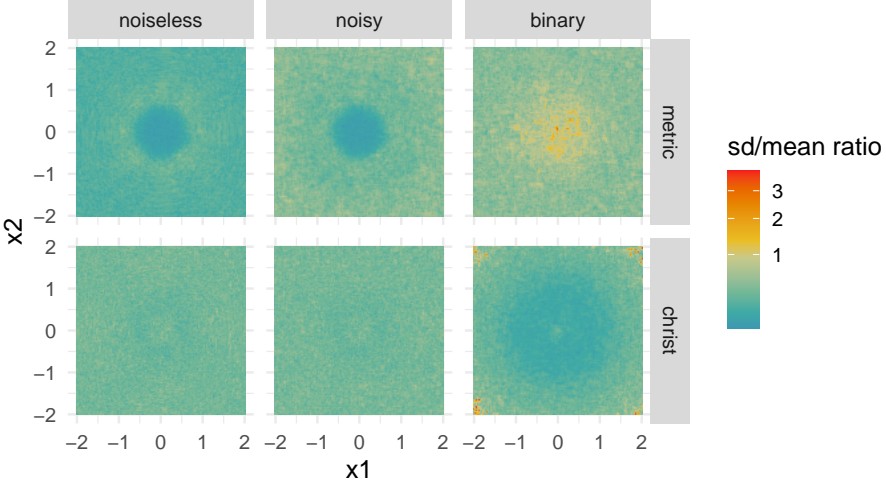

(a) Map over the coordinate chart.

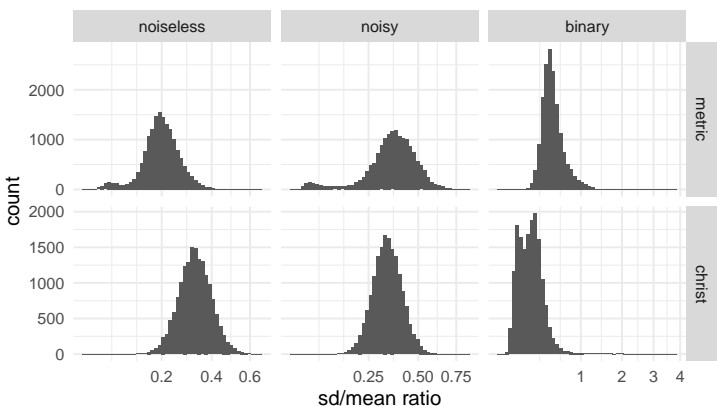

(b) Same as the previous map but histogram.

Figure S4.3: Fluctuation of errors in the estimated tensors among 15 repeats under different random seed for data generation, quantified by ratio of standard deviation to mean at the target points.

## S4.2   Ellipsoid

Our previous example of sphere only considers a model space of constant curvature. To illustrate that the proposed method also applies to manifolds of non-constant curvature, we include in the following example on the ellipsoid.

For simplicity, we resort to spheroids, i.e., ellipsoids of revolution characterized by

$$\left\{ \left(x^1, x^2, x^3\right), (x^i)^2 c_i = 1 \right\} \subset \mathbb{R}^3$$

for some constant $c_1, c_2, c_3 > 0$ with $c_1 = c_2$. It can be parameterized by

$$x^1(\beta, \lambda) = a \cos \beta \cos \lambda, \quad x^2(\beta, \lambda) = a \cos \beta \sin \lambda, \quad x^3(\beta, \lambda) = b \sin \beta$$

for $-\pi/2 < \beta < \pi/2$ and $-\pi < \lambda < \pi$, where $a \geq b > 0$ are the semi-major and semi-minor axis, $\beta$ is the parametric latitude, and $\lambda$ is the longitude. The parametric latitude is slightly different to the usual geodetic latitude $\varphi$, they are related by $\tan \beta = (1 - f) \tan \varphi$, where $f = (a - b)/a$ is the flattening. In the following we use the geodetic latitude unless otherwise specified. We write both latitude and longitude in radian. The Riemannian metric on the spheroid is

$$\frac{a^4 b^4}{\left(a^2 \cos^2 \varphi + b^2 \sin^2 \varphi\right)^3} d\varphi^2 + \frac{a^4}{a^2 + b^2 \tan^2 \varphi} d\lambda^2.$$

We modified the codes in the R library `geosphere` (Hijmans, 2022) to use the C++ library of `GeographicLib` (Karney, 2013) to compute geodesic distance on spheroids.

In a setup similar to the sphere example, we take $10^6$ noisy continuous similarity measures on a spheroid with $a = 2, b = 1$, the estimated Riemannian metric and geodesic curves are shown in Figure S4.4.

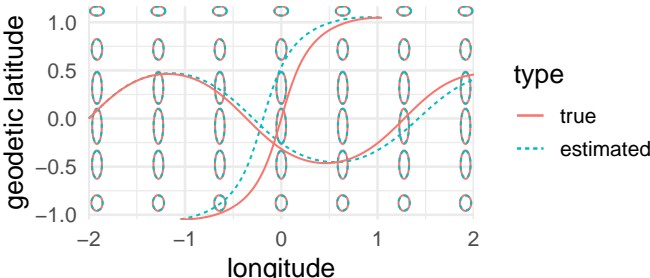

Figure S4.4: The estimated metric and geodesics on a 2-dimensional spheroid with semimajor axis of 2 and flattening of 0.5. The "horizontal" curve is computed starting from $(-2, 0)$ with initial velocity $(1, 1)$, while the "vertical" one between points $(-\pi/3, -\pi/3)$ and $(\pi/3, \pi/3)$.

## S4.3  The Double Spirals

Define a class of spiral functions as $\mathcal{S} : \mathbb{R} \to \mathbb{R}^2$ with

$$t \mapsto (\cos(5t + \phi),\ t\sin(5t + \phi)).$$

The underlying spirals for class $\mathcal{A}$ and $\mathcal{B}$ are $\mathcal{S}_A = \mathcal{S}(\cdot, \phi = 0)$ and $\mathcal{S}_B = \mathcal{S}(\cdot, \phi = \pi)$ respectively. For endpoints, we first generate points $\mathring{X}_m$ independently and uniformly on $\mathcal{S}_A$ or $\mathcal{S}_B$, then the endpoints are generated following $\mathring{X}_m + 0.15 Z_m$ for i.i.d. standard Gaussian random variables $Z_m$, $m = 1, \ldots, 70000$. Provided with those candidate endpoints, we pair them to form relative comparison subject to the restriction that $|X_{u0}^i - X_{uj}^i| \leq 0.35$ for $i, j = 1, 2$, $n = 1, \ldots, N$. The responses $Y_u$ are then generated based on the class of involving endpoints by their corresponding $\mathring{X}$ on the spirals.

For estimation, we used a larger local neighborhood $\mathcal{U}_p = \left\{ (x^1, x^2) : |x^i - p^i| \leq h \text{ for } i = 1, 2 \right\}$ with $h = \pi/2$ and weights $w_u = 1_{\{X_{u0}, X_{u1}, X_{u2} \in \mathcal{U}_p\}}$ for $u = 1, \ldots, N$ to avoid degenerate estimates.

Note that different starting points and initial velocities will generate different geodesics, not all resembling a spiral, as shown in Figure S4.5.

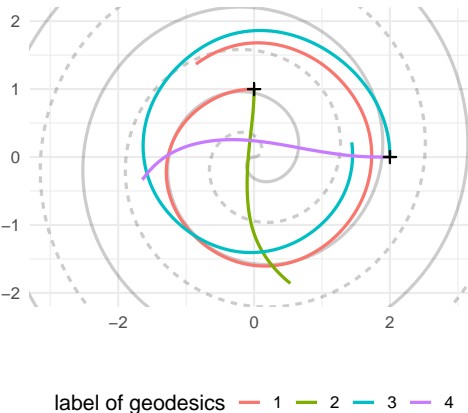

label of geodesics — 1 — 2 — 3 — 4

Figure S4.5: Geodesics with different starting points and initial velocities under estimated metric, crosses indicate starting points.

## S4.4  NYC Taxi Trips

We focus on the 8,809,982 sensible records between 7 a.m. to 10 a.m. on business days from May to September (summer months to hopefully avoid snow) of 2015 in New York City areas other than the Staten Island. Sensible in terms of GPS coordinates not falling in to the rivers, travel time not being several seconds, and that inferred traveling speed is not 120 mph, and e.t.c. We measure the cost to travel $Y_u$ by the squared trip duration (instead of the trip

distance). For each target location $p$, estimation was computed using trips among the $M \leq 5 \times 10^4$ closest pickup/dropoff endpoints in the neighborhood $\mathcal{U}_p = \{(x^1, x^2) : |x^i - p^i| \leq 5 \text{ kilometers for } i = 1, 2\}$, and weights given by $h = 2.5$ kilometers with the kernel $K$ being the density function of the standard normal distribution.

## S4.5  The MNIST Example

The dimension reduction is computed using R package `dimRed` (Kraemer et al., 2018). The Wasserstein distance and optimal transport are computed using package `transport` (Schuhmacher et al., 2022). To show image transitions, weighted average is adopted to approximate the inverse of the tSNE embedding so as to map the trajectories back to image space, similar to, e.g., equation (3.9) of Chen and Müller (2012), but with Gaussian kernel and a sufficiently small bandwidth. To compute kNN graph and find shortest path along the graph, we use `FNN` (Beygelzimer et al., 2022) and `igraph` (Csardi and Nepusz, 2006).

To simplify computation, we only embed the first $3 \times 10^4$ images (half of the entire data), and the resulting embedding coordinates were scaled (i.e., centered by the mean and divided by standard deviation). We generated $N = 10^5$ comparison by selecting nearby points in the embedded space subject to $\|X_{u0} - X_{u1}\|_\infty \leq 0.75$, whose response $Y_u$ were computed based on the same-digit-or-not indicator and 2–Wasserstein distance between the corresponding images:

$$dist(X_{u0}, X_{u0}) = C\, dist_{wass}(\text{pic}_{u0}, \text{pic}_{u1}) + \mathbb{1}_{\{\text{lbl}_{u0} \neq \text{lbl}_{u1}\}},$$

where for $u = 1, \ldots, N$,

- $X_{u0}, X_{u1} \in \mathbb{R}^2$ are coordinates in the embedded space;

- $\text{pic}_{u0}$ and $\text{pic}_{u0}$ are the $28 \times 28$ grey scale images;

- $\text{lbl}_{u0}$ and $\text{lbl}_{u1}$ are the image labels (0–9);

- $dist_{wass}(\cdot, \cdot)$ is the 2–Wasserstein distance treating images as 2-dimensional probability distributions;

- $\mathbb{1}_{\{\text{event}\}}$ is the indicator for whether the event is true (1) or false (0).

We multiply the Wasserstein distance by $C = 4$ to balance the magnitude of the two summands, otherwise the later could be overly dominating.

As for the graphs involved, the neighborhood graph on the embedded space (i.e., on the chart) is constructed by connecting all points together with $\|X_{u0} - X_{u1}\|_\infty \leq 0.1$, and the edges are weighted by to the previously mentioned distance. Again, the $\|\cdot\|_\infty$ is computed on the 2-dimensional chart. In total there were 1,788,166 edges for this graph. We did not use the graph based on the training data of the proposed method ($\|X_{u0} - X_{u1}\|_\infty \leq 0.75$) because that led to very long edge and abrupt graph path. Another graph we used is the kNN

graph in the raw Euclidean space $\mathbb{R}^{28 \times 28}$. We used $k = 5$ and there were 75,003 edges weighted by Euclidean distance of the images.

Estimation was drawn under model (3.2) with squared loss $Q(y, \mu) = (y - \mu)^2$ and the identity link. We included the intercept term ($\beta^{(0)}$ in (3.6)) to capture intrinsic variation. Since the dimensional reduction embedding map is not necessarily an injection, so that different images with non-zero similarity measures could share identical coordinates in the embedded space. Figure S4.6a shows the estimated intercepts, which is larger among class boundaries, coherent to a greater variation in the similarity measure. Those few blank pixels indicate failure to obtain positive definite metric due to lack of local observations, which are alleviated by averaging neighboring estimated values.

For each target location $p$, estimation was computed using comparisons in the neighborhood $\mathcal{U}_p = \{(x^1, x^2) : |x^i - p^i| \leq 0.85, i = 1, 2\}$, and weights given by $h = 0.275$ with the kernel $K$ being the density function of the standard normal distribution. We also dropped the terms for Christoffel symbols from (3.6) for better numeric stability. Consequently, the estimated Christoffel symbols were computed by numeric differential following the definition in Section 2. Results are similar if we include the Christoffel symbol terms in the linear predictor, but less stable and requires larger bandwidth $h$.

Figure S4.6b shows the cost ellipses for addition visualization.

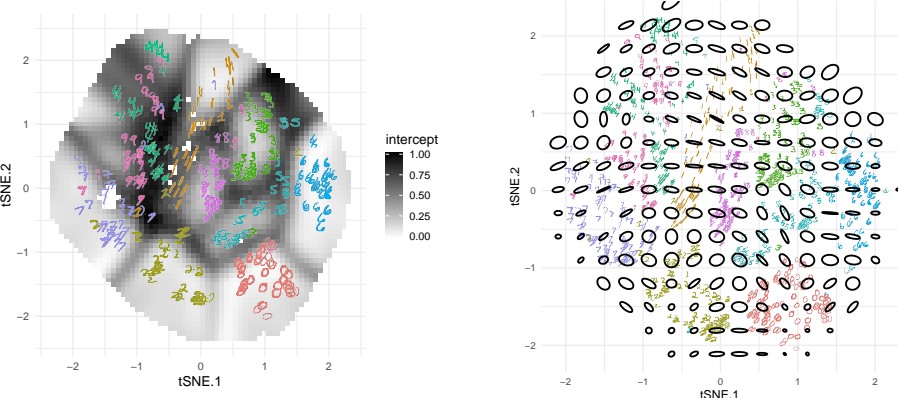

(a) Estimated intercept reflecting intrinsic local variation of the similarity measure.

(b) The cost ellipses of estimated metric.

Figure S4.6: More figures for the induced geometry by adding Wasserstein distance and same-digit-or-not indicator.

Notably, the proposal also work supplied with binary similarity measures using only the same-digit-or-not indicator (i.e., setting $C = 0$ to remove the Wasserstein distance), and retains the "fewer label switching" tendency as illustrated in Figure S4.7. We see this as an real data example for binary responses.

Estimation were now drawn under model (3.3) with loss (3.11) and the logit link. We would also like to remark that not all geodesics are different from straight lines on the chart, and it is not guaranteed that geodesic must travel within the same class whenever possible, since its travel is jointly determined by the metric and the displacement of the endpoints.

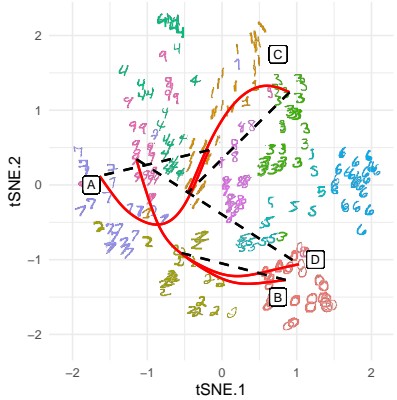

(a) The geodesic curves (solid red) and straight lines (dashed black).

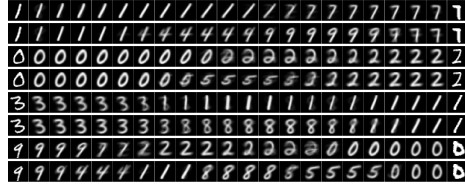

(b) Image transitions (per row) corresponding to bows A, B, C, and D in panel (a). Every 2 rows correspond to one pair of start and end images, where the first and second rows follow geodesics and the straight lines respectively.

Figure S4.7: Induced geodesics of only the same-digit-or-not indicator.

## S4.6 Stability and Positive-definiteness

To ensure the proposal captures underlying geometry of the data space rather than the random artifact, we assess the stability of estimated tensor fields among multiple repeats of the experiments. Since the underlying true tensor fields are either difficult to compute or unavailable (expect for the round metric example discussed in Subsubsection S4.1.2), we assess the fluctuation of the estimates directly. Training data in each repeat were generated under different random seed for the simulated experiments. For the real data experiments, we resampled the data by random dropping out 10% of available similarity measures during each repeat to introduce variation to the training process. Let $\hat{G}_{(i)}(p)$ denotes the estimated metric tensor at point $p$ during the $i$-th repeat. Compute the sample standard deviation $\sigma(p)$ of the relative deviation from mean, i.e., standard deviation of $\left\|\hat{G}_{(1)}(p) - \bar{G}(p)\right\|_F / \left\|\bar{G}(p)\right\|_F, \ldots, \left\|\hat{G}_{(N^*)}(p) - \bar{G}(p)\right\|_F / \left\|\bar{G}(p)\right\|_F$, where $\bar{G}(p)$ is the (element-wise) average of the estimated tensors, $N^*$ is the number of repeated experiments. Then $\sigma(p)$ serves as a quantification of the sensitivity of estimated tensor subject to data variation. Similar assessment applies to the estimated Christoffel symbol $\hat{\Gamma}$ and intercept $\beta^{(0)}$. Figure S4.8 – Figure S4.10 indicates the proposed method is stable, as the larger fluctuation only appears near the boundary. A smaller relative deviation indicates that

the estimator is less sensitive to specific data, and captures the underlying data space geometry. The increased variation in estimated Christoffel symbols as shown in Figure S4.8a is caused by the noisiest data type of relative comparison.

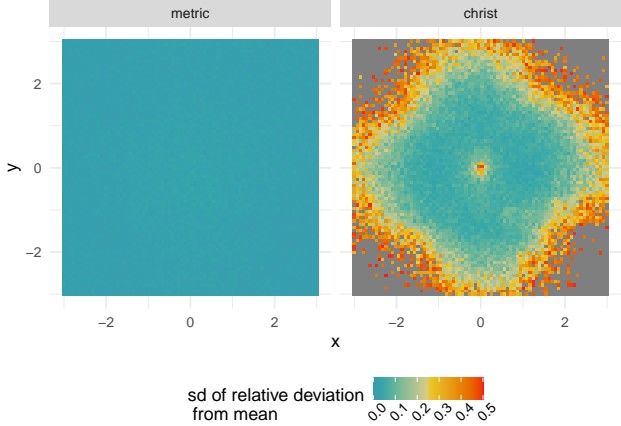

(a) Map over the coordinate chart, values greater than 0.5 grayed out.

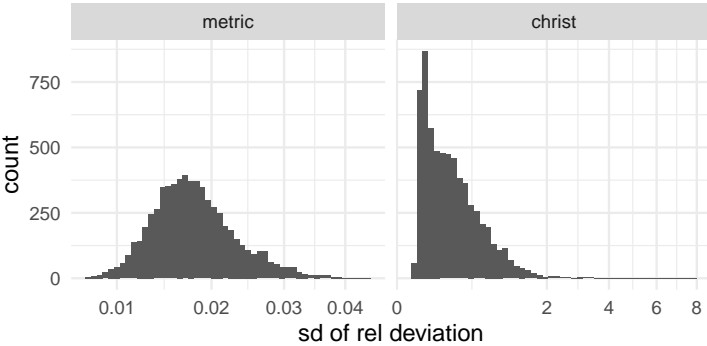

(b) Same as the previous map but histogram.

Figure S4.8: Variation of estimated tensors among 15 repeated experiments for the double spiral example (Subsection 5.2 of main text), quantified by standard deviation of relative deviation to mean.

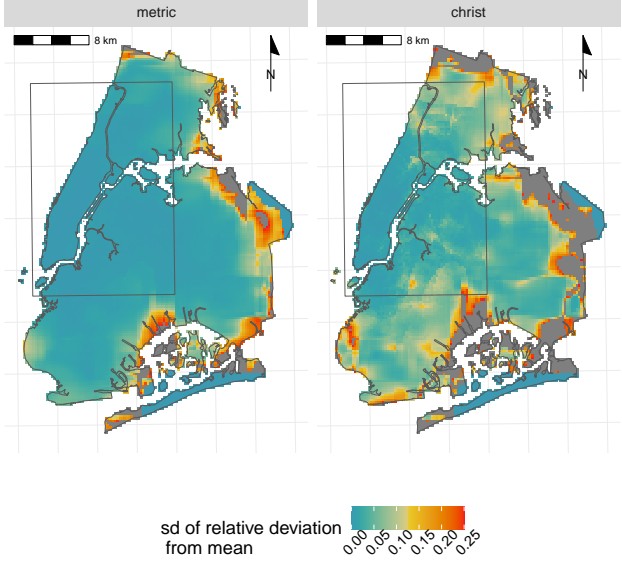

(a) Map over the coordinate chart, values greater than 0.25 grayed out. The rectangle box shows the plotting range of Figure 6.1 in the main text.

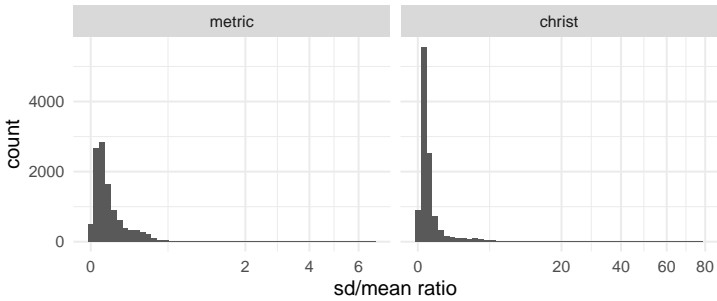

(b) Same as the previous map but histogram.

Figure S4.9: Variation of estimated tensors among 15 resampled data for the New York taxi example (Section 6 of main text), quantified by standard deviation of relative deviation to mean.

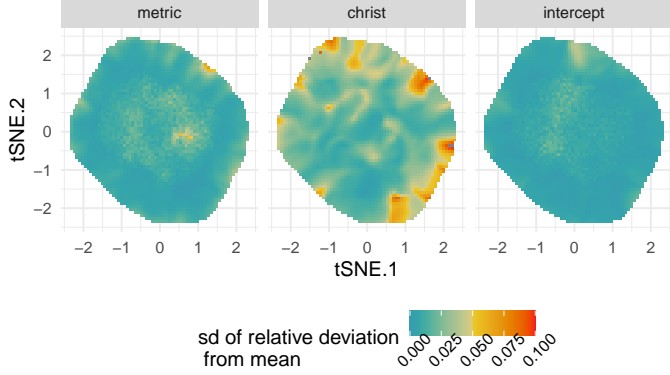

(a) Map over the coordinate chart, values greater than 0.1 grayed out.

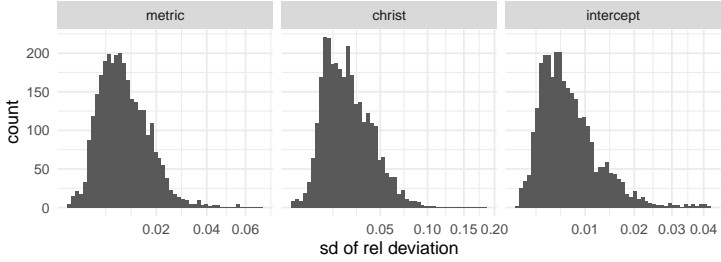

(b) Same as the previous map but histogram.

Figure S4.10: Variation of estimated tensors among 15 resampled data for the MNIST example (Section 7 of main text), quantified by standard deviation of relative deviation to mean.

This resampling process also provide insight to the positive-definiteness of the estimated metric tensors, which was only justified asymptotically. At each targeted point $p$, we examine the proportion of positive definite estimated metrics

$$\frac{1}{N^*} \sum_{i=1}^{N^*} 1\left(\hat{G}_{(i)}(p) \succ 0\right).$$

Figure S4.11 indicates the majority are positive definite except those near the data boundary. We omit the figures for the double spiral example, since only 7 out of 96000 estimated metric tensor fail to be positive definite.

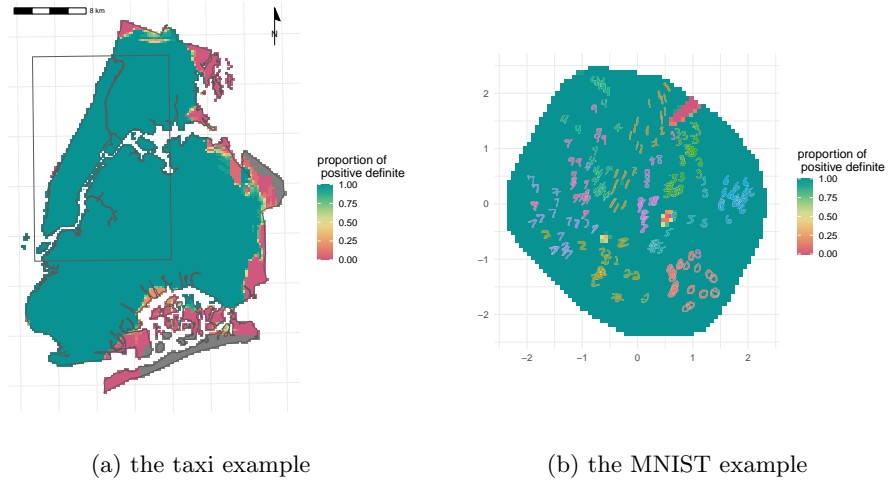

(a) the taxi example            (b) the MNIST example

Figure S4.11: Proportion of positive definite estimated metric tensors among multiple resampled data. Grayed out pixels indicate no estimate due to insufficient neighboring observation.

## S5  Spread of Geodesics

Here we provide proof to the Proposition 3.1 in the main text, which characterizes the distance between geodesics departing from a same starting point. Proposition 3.1 is a result of combining Proposition S5.1 and Proposition S5.2.

**Proposition S5.1** (spread of geodesics). Let $p \in \mathcal{M}$ and $v, w \in T_p\mathcal{M}$ be two tangent vector at $p$. Then the squared distance of separation satisfies Taylor expansion of

$$dist\left(\exp_p(tv), \exp_p(tw)\right)^2 = t^2 \|v - w\|^2 - \frac{1}{3}t^4 \langle R(v, w)w, v \rangle + O(t^5)$$

as $t \to 0$.

Here, $R$ is the $(1, 3)$-*curvature tensor* defined as

$$R(X, Y)Z = \nabla_X \nabla_Y Z - \nabla_Y \nabla_X Z - \nabla_{[X,Y]} Z,$$

where $X, Y, Z$ are vector fields and $[X, Y] = XY - YX$ is the *Lie bracket* (c.f., e.g., Lee, 2018, page 385). Further, the *Riemann curvature tensor* is defined as

$$Rm(X, Y, Z, W) = \langle R(X, Y)Z, W \rangle,$$

where $W$ is also a vector field. Note that $R$ and $Rm$ are both tensor fields, so $\langle R(v, w)w, v \rangle$ (equivalently $Rm(v, w, w, v)$) are those evaluated at $p$, since $v, w \in T_p\mathcal{M}$. See Lee (2018), pp. 196–199 for detail.

However, additional terms are introduced when computing via coordinate charts, as a result of approximating the initial velocities $v$ and $w$.

**Proposition S5.2** (approximation of velocity). For any $p \in \mathcal{M}$, let $v \in T_p\mathcal{M}$ be a tangent vector at $p$ and $\gamma(t) = \exp_p(tv)$ be the geodesic from $p$ with initial velocity $v$. Given any local coordinate chart, write $v = v^i \partial_i$. For $i = 1, \ldots, d$, denote $\delta^i = \delta^i(t) = \gamma^i(t) - \gamma^i(0)$ as the difference in coordinate after traveling $t$ along $\gamma$, we have

$$v^i = t^{-1} \delta^i(t) + \mathscr{R}^i(t),$$

where the remainder is

$$\mathscr{R}^i(t) = \frac{1}{2t} \delta^m \delta^n \Gamma^i_{mn} + \frac{1}{6t} \delta^m \delta^n \delta^l \left( \Gamma^k_{mn} \Gamma^i_{kl} + \partial_l \Gamma^i_{mn} \right) + O(t^3) \qquad \text{(S5.1)}$$

as $t \to 0$, where $\Gamma$ and $\partial\Gamma$ denote the the Christoffel symbols and their derivatives at $p$.

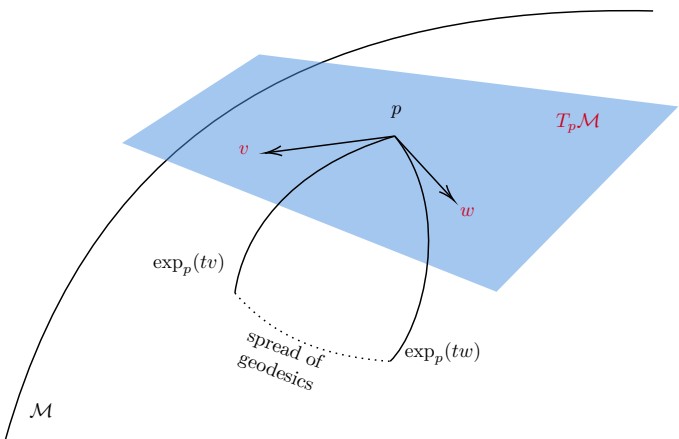

Figure S5.1: A visualization for the spread of geodesics as in Proposition 3.1. A tangent space (blue plane) and tangent vectors are annotated in red.

## S5.1 Proofs

*Proof of Proposition S5.1.* Similar results can be found at Proposition 2.7 of do Carmo (1992), Proposition 5.4, of (Lang, 1999, IX, §5). We use the form presented by Meyer (1989). In the following, we reproduce the proof to the equation (9) of Meyer (1989) with some additional clarification.

Let $\gamma_0(s) = \exp_p(sv)$ and $\gamma_1(s) = \exp_p(sw)$, define a family of curves

$$V(s,t) = \exp_{\gamma_0(s)}\left(t\exp_{\gamma_0(s)}^{-1}\gamma_1(s)\right),$$

so that the curves $V_s : t \mapsto V(s,t)$ are geodesics connecting $\gamma_0(s)$ and $\gamma_1(s)$ (c.f., e.g., proposition 5.19 and equation (10.2) of Lee (2018)), and that $V$ is a variation through geodesics $V_s$. Further, let $T = \partial_t V$, which is a tangent field of velocities. Let $E = \partial_s V$, which is a Jacobi field through geodesics $V_s$ that vanishes at $p$. Denote $H(s) = dist\left(\gamma_0(s), \gamma_1(s)\right)^2 = \|T\|^2|_{s,t}$ for any $t \in [0,1]$, where the "$|_{s,t}$" means to take value at point $V(s,t)$. Then by the chain rules for covariant derivatives (see, e.g. Lee, 2018, chapter 4), we have

$$\frac{d}{ds}H(s) = \frac{d}{ds}\langle T,T\rangle|_{s,t} = 2\langle D_sT, T\rangle|_{s,t},$$

$$\left(\frac{d}{ds}\right)^2 H(s) = 2\left(\langle D_s^2T, T\rangle + \|D_sT\|^2\right)|_{s,t},$$

$$\left(\frac{d}{ds}\right)^3 H(s) = 2\left(\langle D_s^3T, T\rangle + 3\langle D_s^2T, D_sT\rangle\right)|_{s,t},$$

$$\left(\frac{d}{ds}\right)^4 H(s) = 2\left(\langle D_s^4T, T\rangle + 3\left\|D_s^2T\right\| + 4\langle D_s^3T, D_sT\rangle\right)|_{s,t}.$$

Note that $V_0 = p$ for all $t$, so that $T|_{s=0,t} = 0$ for all $t$, hence $H'(0) = 0$.

Note that $V_s : t \mapsto V(s,t)$, $s \mapsto V(s,0)$ and $s \mapsto V(s,1)$ are geodesics; thus $D_tT = 0$ for all $t$, $D_sE|_{s,t=0} = 0$ and $D_sE|_{s,t=1} = 0$ for all $s$. In addition, by lemma 6.2 of Lee (2018), $D_sT = D_tE$.

By Jacobi equation, $D_t^2E + R(E,T)T = 0$ for all $s$, which implies $D_t^2E|_{s=0} = 0$ since $T|_{s=0} = 0$. This means the vector field $t \mapsto E|_{s=0,t}$ at $p$ is linear in $t$, together with $E|_{s=0,t=0} = v$ and $E|_{s=0,t=1} = w$, we can write

$$E|_{s=0,t} = v + t(w-v)$$

for $t \in [0,1]$. Therefore $D_sT|_{s=0,t} = D_tE|_{s=0,t} = w - v$, which implies $H''(0) = 2\|v-w\|^2$.

Proceeding to the third order derivatives, observe that

$$H'''(0) = 6\langle D_s^2T, D_sT\rangle|_{s=0,t},$$

and by proposition 7.5 of Lee (2018), $D_s^2T = D_sD_tE = D_tD_sE + R(E,T)E$, thus it suffices to show

$$D_sE|_{s=0,t} = 0, \text{ for all } t, \tag{S5.2}$$

in order to argue $H'''(0) = 0$. Since it is known that $D_sE|_{s=0,t=0} = 0 = D_sE|_{s=0,t=1}$, it suffices to consider its derivative for (S5.2). Use proposition 7.5 of Lee (2018) repeatedly, we have

$$D_t^2 D_s E|_{s=0,t}$$
$$= D_t D_s D_t E|_{s=0,t} + D_t\left(R(T,E)E\right)|_{s=0,t}$$
$$= D_t D_s^2 T|_{s=0,t}$$
$$= \left(D_s D_t D_s T - R(E,T)(D_s T)\right)|_{s=0,t}$$
$$= D_s D_t D_s T|_{s=0,t}$$
$$= D_s\left(D_s D_t T - R(E,T)T\right)|_{s=0,t}$$
$$= D_s\left(R(T,E)T\right)|_{s=0,t},$$

where the last equation is due to $D_t T = 0$. Further, by chain rule of covariant derivative (c.f. e.g. proposition 4.15 of Lee (2018)),

$$D_s\left(R(T,E)T\right) = \left(\nabla_E R\right)(T,E)T + R(D_s T,E)T + R(T,D_s E)T + R(T,E)D_s T,$$

which equals to zero at $s=0, t$ since $T|_{s=0,t} = 0$ for all $t$. Hence $t \mapsto D_s E|_{s=0,t}$ is also a linear vector field, implying (S5.2) and subsequently $H'''(0) = 0$.

For the fourth order derivative, note that (S5.2) also implies that $D_t D_s E|_{s=0,t} = 0$ and that $D_s^2 T|_{s=0,t} = 0$ for all $t$. Therefore,

$$H^{(4)}(0) = 8\left\langle D_s^3 T, D_s T\right\rangle|_{s=0,t}.$$

Further,

$$D_s^3 T = D_s^2 D_t E$$
$$= D_s\left(D_t D_s E + R(E,T)E\right)$$
$$= D_s D_t D_s E + \left(\nabla_E R\right)(E,T)E + R(D_s E,T)E + R(E,D_s T)E + R(E,T)(D_s E),$$

so $D_s^3 T|_{s=0,t} = \left(D_s D_t D_s E + R(E,D_s T)E\right)|_{s=0,t}$. Thus,

$$\left\langle D_s^3 T, D_s T\right\rangle|_{s=0,t} = \left(\left\langle D_s D_t D_s E, D_s T\right\rangle + \left\langle R(E,D_s T)E, D_s T\right\rangle\right)|_{s=0,t}.$$

Recall at $s=0$, $D_s T|_{s=0,t} = D_t E|_{s=0,t} = w - v$, therefore

$$\left\langle R(E,D_s T)E, D_s T\right\rangle|_{s=0,t}$$
$$= Rm(E,D_s T,E,D_s T)|_{s=0,t}$$
$$= Rm(v,w-v,v,w-v) + tRm(v,w-v,w-v,w-v)$$
$$= Rm(v,w,v,w).$$

Further,

$$\left\langle D_s D_t D_s E, D_s T\right\rangle|_{s=0,t}$$
$$= \left(D_s\left\langle D_t D_s E, D_s T\right\rangle - \left\langle D_t D_s E, D_s^2 T\right\rangle\right)|_{s=0,t}$$
$$= D_s\left\langle D_t D_s E, D_s T\right\rangle|_{s=0,t}$$
$$= D_s D_t\left\langle D_s E, D_s T\right\rangle|_{s=0,t} - D_s\left\langle D_s E, D_t^2 E\right\rangle|_{s=0,t},$$

where the second term in the last line vanishes since $D_t^2 E|_{s=0,t} = 0$ and $D_s E|_{s=0,t} = 0$. Moreover, since the Levi–Civita connection is torsion free, we have

$$\langle D_s D_t D_s E, D_s T \rangle |_{s=0,t} = D_s D_t \langle D_s E, D_s T \rangle |_{s=0,t} = D_t D_s \langle D_s E, D_s T \rangle |_{s=0,t},$$

which should be irrelevant to $t$, so that $D_s \langle D_s E, D_s T \rangle |_{s=0,t}$ is linear in $t$. Yet

$$D_s \langle D_s E, D_s T \rangle = \langle D_s^2 E, D_s T \rangle + \langle D_s E, D_s^2 T \rangle,$$

which vanishes at $s = 0$ and $t = 0, 1$. Hence $D_s \langle D_s E, D_s T \rangle |_{s=0,t} = 0$ for all $t \in [0,1]$. Combining those with the symmetries of Riemann curvature tensor leads to the desired expansion. $\square$

*Proof of Proposition S5.2.* Under the coordinate chart, we can write the geodesic curve as $\gamma : t \mapsto \left( \gamma^1(t), \ldots, \gamma^d(t) \right)$ for some smooth function $\gamma^1, \ldots, \gamma^d$. Then for any $i = 1, \ldots, d$, univariate Taylor expansion provides

$$\gamma^i(t) = \gamma^i(0) + \dot{\gamma}^i(0)t + \frac{1}{2} t^2 \ddot{\gamma}^i(0) + \frac{1}{6} t^3 \dddot{\gamma}^i(0) + O(t^4)$$

as $t \to 0$, where $\dot{\gamma}^i$, $\ddot{\gamma}^i$, and $\dddot{\gamma}^i$ are the first, second, and third order derivative of $\gamma^i$ w.r.t. $t$. Note that the first derivative $\dot{\gamma}^i(0) = v^i$, and the geodesic equation and its derivative give

$$\ddot{\gamma}^i(0) = -v^m v^n \Gamma_{mn}^i,$$
$$\dddot{\gamma}^i(0) = v^m v^n v^l \left( 2\Gamma_{mn}^k \Gamma_{kl}^i - \partial_l \Gamma_{mn}^i \right).$$

Plugging into the initial Taylor expansion gives the desired result. $\square$

*Proof of Proposition 3.1 in the maintext.* By Proposition S5.1 and Proposition S5.2, as $t \to 0$, we have

$$\begin{aligned}
& t^2 \| v - w \|^2 \\
&= \left( \delta_{0-1}^i + t\mathcal{R}_0^i(t) - t\mathcal{R}_1^i(t) \right) G_{ij} \left( \delta_{0-1}^j + t\mathcal{R}_0^j(t) - t\mathcal{R}_1^j(t) \right) \\
&= \delta_{0-1}^i \delta_{0-1}^j G_{ij} + 2t \delta_{0-1}^i \left( \mathcal{R}_0^j(t) - \mathcal{R}_1^j(t) \right) G_{ij} + O(t^4) \\
&= \delta_{0-1}^i \delta_{0-1}^j G_{ij} + \delta_{0-1}^i \left( \delta_0^k \delta_0^l - \delta_1^k \delta_1^l \right) \left( \Gamma_{kl}^j G_{ij} \right) + O(t^4),
\end{aligned}$$

where

$$\mathcal{R}_a^i(t) = \frac{1}{2t} \delta_a^m \delta_a^n \Gamma_{mn}^i + \frac{1}{6t} \delta_a^m \delta_a^n \delta_a^l \left( \Gamma_{mn}^k \Gamma_{kl}^i + \partial_l \Gamma_{mn}^i \right) + O(t^3)$$

for $a = 0, 1$, similar to (S5.1). Note that $\delta_0^i = \delta_0^i(t) = O(t)$, it suffices to keep only the first term in the $\mathcal{R}_a^j(t)$, which is $O(t)$. $\square$

# S6    Asymptotic of the Estimated Metric Tensor

Now we discuss the variance and bias of the estimated metric tensors. For simplicity, use the squared loss $Q(\mu, y) = (\mu - y)^2$, the identity link $g(\mu) = \mu$, and exclude the intercept $\beta^{(0)}$ and the terms $\beta_{ijk}^{(2)}$ for derivative. Given a suitable order of the indices $i, j$, we rewrite (3.6) into matrix form. Denote

$$\mathbf{D}_u = \left( \delta_{u,0-1}^1 \delta_{u,0-1}^1, \ldots, \delta_{u,0-1}^i \delta_{u,0-1}^j, \ldots, \delta_{u,0-1}^d \delta_{u,0-1}^d \right)^T,$$

$$\boldsymbol{\beta} = \left( \beta_{11}^{(1)}, \ldots, \beta_{ij}^{(1)}, \ldots, \beta_{dd}^{(1)} \right)^T,$$

then the linear predictor $\eta_n = \mathbf{D}_u^T \boldsymbol{\beta}$. Further, write

$$\mathbf{D} = (\mathbf{D}_1, \ldots, \mathbf{D}_N)^T, \quad \boldsymbol{Y} = (Y_1, \ldots, Y_N)^T, \quad \mathbf{W} = \text{diag}(w_1, \ldots, w_N),$$

so the loss (3.7) becomes

$$(\boldsymbol{Y} - \mathbf{D}\boldsymbol{\beta})^T \mathbf{W} (\boldsymbol{Y} - \mathbf{D}\boldsymbol{\beta}), \tag{S6.1}$$

whose minimizer is $\hat{\boldsymbol{\beta}} = \left( \mathbf{D}^T \mathbf{W} \mathbf{D} \right)^{-1} \mathbf{D}^T \mathbf{W} \boldsymbol{Y}$. Therefore

$$\text{bias} \left( \hat{\boldsymbol{\beta}} | \mathbf{D} \right) = \left( \mathbf{D}^T \mathbf{W} \mathbf{D} \right)^{-1} \mathbf{D}^T \mathbf{W} \boldsymbol{r},$$

$$\text{var} \left( \hat{\boldsymbol{\beta}} | \mathbf{D} \right) = \left( \mathbf{D}^T \mathbf{W} \mathbf{D} \right)^{-1} \mathbf{D}^T \boldsymbol{\Sigma} \mathbf{D} \left( \mathbf{D}^T \mathbf{W} \mathbf{D} \right)^{-1},$$

where

$$\boldsymbol{r} = \left( \mathbb{E} \left( Y_u | \mathbf{D} \right) - \delta_{u,0-1}^i \delta_{u,0-1}^j G_{ij} \right)_{1 \leq n \leq N},$$

$$\boldsymbol{\Sigma} = \text{diag} \left( w_u^2 \, \text{var} \left( Y_u | X_{u0}, X_{u1} \right) \right)_{1 \leq n \leq N}.$$

The assumptions (A1)–(A4) in the main text are reiterated here.

(A1) The joint density of endpoints $X_{u0}, X_{u1}$ is positive and continuously differentiable.

(A2) The functions $G_{ij}, \Gamma_{ij}^k$ are $C^2$-smooth for $i, j, k = 1, \ldots, d$.

(A3) The kernel $K$ in weights (3.10) is symmetric, continuous, and has bounded support.

(A4) $\sup_u \text{var}\left( Y_u | X_{u0}, X_{u1} \right) < \infty$.

**Proposition S6.1.** Denote

$$S_{1N, i_1 i_2 i_3 i_4} = \sum_{u=1}^N w_u \delta_{u,0-1}^{i_1} \delta_{u,0-1}^{i_2} \delta_{u,0-1}^{i_3} \delta_{u,0-1}^{i_4},$$

$$S_{2N, i_1 i_2 i_3 i_4} = \sum_{u=1}^N w_u^2 \, \text{var} \left( Y_u | X_{u0}, X_{u1} \right) \delta_{u,0-1}^{i_1} \delta_{u,0-1}^{i_2} \delta_{u,0-1}^{i_3} \delta_{u,0-1}^{i_4},$$

$$S_{3N, i_1 i_2} = \sum_{u=1}^N w_u \delta_{u,0-1}^{i_1} \delta_{u,0-1}^{i_2} R_u,$$

where
$$R_u = \sum_{1 \leq k,l,m,r \leq d} \delta_{u,0-1}^m \left( \delta_{n0}^k \delta_{n0}^l - \delta_{n1}^k \delta_{n1}^l \right) \Gamma_{kl}^r G_{mr}.$$

Under (A1), (A2), (A3), and (A4), and suppose that $h \to 0$ and $Nh^{2d} \to \infty$, then

$$\mathbb{E}S_{1N,i_1i_2i_3i_4} = O\left(Nh^4\right), \quad \text{var} \, S_{1N,i_1i_2i_3i_4} = O\left(Nh^{8-2d}\right),$$
$$\mathbb{E}S_{2N,i_1i_2i_3i_4} = O\left(Nh^{4-2d}\right), \quad \text{var} \, S_{2N,i_1i_2i_3i_4} = O\left(Nh^{8-6d}\right),$$
$$\mathbb{E}S_{3N,i_1i_2i_3i_4} = O\left(Nh^6\right), \quad \text{var} \, S_{3N,i_1i_2i_3i_4} = O\left(Nh^{10-2d}\right),$$

as $h \to 0$ and $N \to \infty$. So

$$S_{1N,i_1i_2i_3i_4} = O_p\left(Nh^4\right), \quad S_{2N,i_1i_2i_3i_4} = O_p\left(Nh^{4-2d}\right).$$

If further $Nh^{2+2d} \to \infty$, then

$$S_{3N,i_1i_2i_3i_4} = O_p\left(Nh^6\right),$$

as $h \to 0$ and $N \to \infty$.

*Proof.* Write

$$U_{u;i_1i_2i_3i_4} = w_u \delta_{u,0-1}^{i_1} \delta_{u,0-1}^{i_2} \delta_{u,0-1}^{i_3} \delta_{u,0-1}^{i_4},$$

so

$$\mathbb{E}U_{u;i_1i_2i_3i_4}$$
$$= \int h^{-2d} \prod_{i=1}^d \left( K\left(\delta_{n0}^i/h\right) K\left(\delta_{n1}^i/h\right) \right) \delta_{n,0-1}^{i_1} \delta_{n,0-1}^{i_2} \delta_{n,0-1}^{i_3} \delta_{n,0-1}^{i_4} \cdot$$
$$f(p^1 + \delta_{n0}^1, \ldots, p^d + \delta_{n1}^d) d\delta_{n0}^1 \ldots d\delta_{n1}^d$$
$$= h^4 \int K\left(s_{u0}^1\right) \cdot \ldots \cdot K\left(s_{u1}^d\right) \left(s_{u0}^{i_1} - s_{u1}^{i_1}\right) \left(s_{u0}^{i_2} - s_{u1}^{i_2}\right) \left(s_{u0}^{i_3} - s_{u1}^{i_3}\right) \left(s_{u0}^{i_4} - s_{u1}^{i_4}\right) \cdot$$
$$\left(f(p^1, \ldots p^d) + o(1)\right) ds_{u0}^1 \ldots ds_{u1}^d$$
$$= O(h^4),$$

where $f$ is the joint density of endpoints $X_{u0}, X_{u1}$, the second last equality is due to change of variables, and the last due to (A1). Similar argument implies

$$\text{var} \, U_{u;i_1i_2i_3i_4} = O\left(h^{8-2d}\right),$$
$$\mathbb{E}w_u U_{u;i_1i_2i_3i_4} = O\left(h^{4-2d}\right),$$
$$\text{var} \, w_u U_{u;i_1i_2i_3i_4} = O\left(h^{8-6d}\right).$$

These rates apply uniformly over $n$, therefore by i.i.d. and that $\text{var} \, Y_u|X_{u0}, X_{u1}$ is uniformly bounded,

$$\mathbb{E}S_{1N,i_1i_2i_3i_4} = O\left(Nh^4\right), \quad \text{var} \, S_{1N,i_1i_2i_3i_4} = O\left(Nh^{8-2d}\right),$$
$$\mathbb{E}S_{2N,i_1i_2i_3i_4} = O\left(Nh^{4-2d}\right), \quad \text{var} \, S_{2N,i_1i_2i_3i_4} = O\left(Nh^{8-6d}\right).$$

Hence

$$S_{1N,i_1i_2i_3i_4} = \mathbb{E}S_{1N,i_1i_2i_3i_4} + O_p\left(\sqrt{\operatorname{var} S_{1N,i_1i_2i_3i_4}}\right) = O_p\left(Nh^4\right),$$

under $h \to 0$ and $Nh^{2d} \to \infty$. Similarly we have results for $S_{2N,i_1i_2i_3i_4}$.

Next, write

$$V_{u;i_1i_2} = w_u \delta^{i_1}_{u,0-1} \delta^{i_2}_{u,0-1} R_u.$$

Note that

$$
\begin{aligned}
&\mathbb{E}V_{u;i_1i_2} \\
&= \int h^{-2d} \prod_{i=1}^{d} \left(K\left(\delta^i_{n0}/h\right) K\left(\delta^i_{n1}/h\right)\right) \delta^{i_1}_{n,0-1} \delta^{i_2}_{n,0-1} \times \\
&\qquad \sum_{1 \le k,l,m,r \le d} \delta^m_{u,0-1} \left(\delta^k_{n0}\delta^l_{n0} - \delta^k_{n1}\delta^l_{n1}\right) F_{klm} \times \\
&\qquad f(p^1 + \delta^1_{n0}, \dots, p^d + \delta^d_{n1}) d\delta^1_{n0} \dots d\delta^d_{n1} \\
&= h^5 \int K\left(s^1_{u0}\right) \cdot \dots \cdot K\left(s^d_{u1}\right) \left(s^{i_1}_{u0} - s^{i_1}_{u1}\right) \left(s^{i_2}_{u0} - s^{i_2}_{u1}\right) \times \\
&\qquad \sum_{1 \le k,l,m,r \le d} \left(s^{i_2}_{u0} - s^{i_2}_{u1}\right) \left(s^k_{u0}s^l_{u0} - s^k_{u1}s^l_{u1}\right) F_{klm} \times \\
&\qquad \left(f(p^1, \dots p^d) + h \sum_{r=1}^{d} \frac{\partial f}{\partial p^r}(p)\left(s^r_{u0} + s^r_{u1}\right) + o(h)\right) ds^1_{u0} \dots ds^d_{u1} \\
&= O(h^6),
\end{aligned}
$$

where $F_{klm} = \Gamma^r_{kl} G_{mr}$. Indeed, since the kernel $K$ is symmetric, and the leading terms in the integrant is of fifth power of $s$, thus with some abuse of notation, $\mathbb{E}V_{u;i_1i_2} = h^5 F \int K(s)s^5\left(f + hO(s)\right) ds = O(h^6)$. Similarly

$$\operatorname{var} V_{u;i_1i_2} = O(h^{10-2d}).$$

The rest of the results for $S_{3N,i_1i_2i_3i_4}$ proceeds analogously to that of $S_{1N,i_1i_2i_3i_4}$ and $S_{2N,i_1i_2i_3i_4}$. □

**Proposition S6.2.** Under the conditions of Proposition S6.1,

$$\operatorname{bias}\left(\hat{\boldsymbol{\beta}}|\mathbf{X}\right) = O_p\left(h^2\right), \quad \operatorname{var}\left(\hat{\boldsymbol{\beta}}|\mathbf{X}\right) = O_p\left(\frac{1}{Nh^{4+2d}}\right),$$

as $h \to 0$ and $Nh^{2+2d} \to \infty$, where $\mathbf{X}$ are the observed endpoints.

*Proof.* Note that $S_{1N;i_1i_2i_3i_4}$ are elements of $\mathbf{D}^T\mathbf{W}\mathbf{D}$, where one pair of $(i_1, i_2)$ index a row while one pair of $(i_3, i_4)$ index a column for $i_1, i_2, i_3, i_4 = 1, \dots, d$. Similarly $S_{2N;i_1i_2i_3i_4}$ are elements of $\mathbf{D}^T\boldsymbol{\Sigma}\mathbf{D}$, and $S_{3N;i_1i_2}$ are elements of $\mathbf{D}^T\mathbf{W}\boldsymbol{r}$ by Proposition 3.1. Applying Proposition S6.1 leads to the result. □