# OpenReview forum: "Estimating Riemannian Metric with Noise-Contaminated Intrinsic Distance"
_NeurIPS.cc/2023/Conference — NeurIPS 2023 poster_

### Official Review · Reviewer_uGdp · 2023-06-16

**Soundness:** 4 excellent
**Presentation:** 3 good
**Contribution:** 3 good
**Rating:** 7
**Confidence:** 4

**Summary:**

The paper presents a novel mechanism for learning a Riemannian metric from distance observations. This is important for applications where relative observations are available (e.g. "objects x1 and x2 are different, while x2 and x3 are similar"), such as perception studies. The approach is based on local regression from which a Riemannian metric is deduced from a Taylor expansion of the geodesic distance. Limited experimental results demonstrate feasibility.

**Strengths:**

* The paper approaches an important question that is highly understudied in the machine-learning community.
* The proposed Riemannian metric estimator is new and sufficiently simple to be practically useful.
* It is neat that the estimator also comes with a direct estimate of the Cristoffel symbols as these can otherwise be tedious to compute.
* The examples given throughout the paper are instructive and provide a nice assistance to the reader.
* The illustrative examples are instructive.

**Weaknesses:**

* The approach seems to be difficult to scale to higher dimensional data. I don't think this is a significant problem as the current application areas where distance observations are available often revolve around low-dimensional observations, e.g. in perception studies in psychology.
* The computational procedure seems to rely on a discretization of the input space, which will only work in low-dimensional cases.
* The kernel smoothing used to produce the Riemannian metric (eq. 3.10) is rather ad hoc. It would have been nice if this was more closely tied to the local regression. That being said, the approach basically follows established procedures, see e.g. [10].

### Related work
* I am not fond of the phrasing of lines 37-39, which suggests that the proposed work is the first time that Riemannian metrics are learned. The cited work of Lebanon [14] and Hauberg et al. [10] follows the same philosophy as the present paper, so I think a rephrasing would be appropriate.
* The authors are not the first to learn metrics from observations of distances. I think it would be good to at least acknowledge the vast literature on multi-dimensional scaling, which does not learn a Riemannian metric but learns from distances. Perhaps closer to the present work is the paper "Isometric Gaussian Process Latent Variable Model for Dissimilarity Data" (Jørgensen et al., ICML 2021) which learns a low-dimensional manifold and its Riemannian metric from noisy distance observations. The taken approach is quite dissimilar from the present paper, but the ambition is similar.

### Minor things
* Typo in line 219: "without or without" --- I suppose this should be "with or without".

**Questions:**

* What does the $\circ$ operator denote in Eq. 2.1? As I read the equation, all quantities are scalars, so I do not think that this denotes a Hadamard product, but this is nonetheless my best guess.
* I do not quite understand Eq. 3.2: Are there no further restrictions in $\epsilon_n$? If so does that not mean that $P(Y < 0) > 0$, which is rather odd for distance observations which must be strictly non-negative.
* There has been quite extensive work on pull-back metrics in autoencoder-like models (see e.g. "Latent Space Oddity: on the Curvature of Deep Generative Models", Arvanitidis et al., ICLR 2018). These can be viewed as smooth interpolations of local linear regressions since the metric is formed by Jacobian matrices. Do you see links between this line of work on the present paper?

**Limitations:**

The main paper only makes a brief mention of the limitations, while a more extensive discussion is given in the supplements. I appreciate the supplementary discussion but would encourage the authors to move this to the main paper to increase openness regarding limitations.

---

> ### Author Rebuttal · Authors · 2023-08-10
>
> For concerns regarding scalability to high-dim data, we kindly refer the reviewer to our general point 1. We also thank the reviewer for pointing out many related works, as discussed in our general point 3, we will add citations accordingly. We also discuss the positivity question in the general point 8. We also fix the typo in line 219 (for a camera-ready version hopefully), thank you for the careful inspection.
>
> The $\circ$ operator in eq. (2.1) is function composition, i.e., $f \circ g (t) = f(g(t))$. Adopted to avoid too many brackets especially in our case with many sub/superscripts.
>
> ## Ad hoc kernel smoothing in eq. (3.10)
> The objective (3.7) and weights in equation (3.10) are motivated by the Taylor expansion (3.5) similarly to that in the local polynomial regression (e.g., section 5.4.3 of \[6\]). The weights (3.10) are relatively standard, treating the $(\delta_{u0}^1, \dots, \delta_{u0}^d, \delta_{u1}^1, \dots, \delta_{u1}^d)$ as a $2d$-array. This method has a solid theoretical foundation as established in our Proposition 4.1. One key difference between our proposal to that in \[10\] is that the weights are applied differently. We estimate the Riemannian metric tensor directly and the weights (3.10) are involved in the objective function (3.7), while the method in \[10\] constructs the metric tensor by smoothing multiple local distance metric (see eq. (6) in \[10\]), in a similar manner as our post-smoothing step (section S3 in the Supplement).
>
> ## Potentially misleading claim in lines 37-39
> Lines 37-39 (and this paragraph spanning lines 34-42) emphasize that it is the *Riemannian metric* that we are targeting which is different to the *distance metric* as in the classic metric learning. It does not imply that this is the first time Riemannian metrics are learned, as we have referred to multiple references (as in lines 22-26) including those you kindly pointed out.
>
> ## Positivity
> While the estimated Riemannian metric in a finite sample is not guaranteed to be positive definite, our asymptotic theory (proposition 4.1) shows that the estimated metric will be positive definite with probability tending to 1 given sufficient observations (near the point of estimation). In our experiments we rarely had estimates that were not positive definite and we did not encounter serious non-positive issues as shown in the last part of section S4.5 (page 15) and figure S4.10 of the Supplement. One could in principle adopt constraint optimization to enforce positive definiteness as that in the metric learning, but for simplicity we chose to employ an unrestricted algorithm that is more computationally efficient.
> Model (3.2) is general in that it can include both positive and negative distances, but in practice only positive distances are observed and our model is applied on these more intuitive setups. One could switch the link function and/or the loss function in a similar manner as in local quasi-likelihood [(Fan, Heckman, and Wand 1995)](https://www.tandfonline.com/doi/abs/10.1080/01621459.1995.10476496) to ensure strictly non-negative distance. But again, we chose to present the simplest model here.
>
> ## Link to works on pull-back metrics in autoencoder-like models
> The pull-back approach tackles a similar problem recovering data space geometry. It utilizes the Jacobian matrix of some smooth mapping $f: \mathcal Z \to \mathcal X$ to pull the metric on $\mathcal X$ back to $\mathcal Z$. Typically the mapping $f$ is a generator/encoder, while $\mathcal Z$ and $\mathcal X$ are some latent space and the data space respectively. It is usually assumed that the metric $\mathcal M_x$ of input space is readily available, e.g., assumed to be Euclidean ([Arvanitidis, Hansen, and Hauberg 2018](https://openreview.net/forum?id=SJzRZ-WCZ)), or estimated via Riemannian metric learning ([Arvanitidis, Hauberg, and Schölkopf 2020](http://arxiv.org/abs/2008.00565)).
> Our proposal can provide the metric of latent space directly based on similarity measures in the data space in a manner similar to our MNIST example. The major difference is that our method treats the low-dim embedding as a coordinate chart, which is a stricter assumption: the pull-back metric only requires smooth mappings between the data and latent space (e.g., generator or encoder).

---

> > ### Comment · Reviewer_uGdp · 2023-08-15
> > **Thanks for the follow-up**
> >
> > I appreciate the rebuttal. In particular, I had not seen the supplementary discussions which were pointed out. I found these helpful.
> >
> > I have bumped my score a bit to reflect that I think the paper has merit and it should be published.
> >
> > I think issues regarding scaling to higher dimensions remain, but given the limited work done in this field, I can accept that early papers are incomplete.

---

> > > ### Author Response · Authors · 2023-08-16
> > >
> > > Thank you for your follow-up and appreciation of our work!

---

### Official Review · Reviewer_9sCn · 2023-06-29

**Soundness:** 3 good
**Presentation:** 2 fair
**Contribution:** 2 fair
**Rating:** 4
**Confidence:** 3

**Summary:**

The paper aims to extend metric and manifold learning by learning Riemannian metrics from functions of the observed data that are not related to an embedding space metric as is the usual case. Assuming there is an underlying Riemannian metric structure and that the observed dissimilarity is a known function of this structure, the authors show how a Riemannian metric can be learned from dense data. The method is tested on trip time data in New York and MNIST represented on a 2d manifold.


**Strengths:**

- methods for estimating a Riemannian metric from dissimilarity measures are derived
- in principle, it can be a good idea relying on other distances than embedding space distances to learn geometric structure

**Weaknesses:**

- learning Riemannian metrics has a long history in the literature, although not in this exact form
- I am unsure of the usefulness of the method:
-- are there any guarantees that a Riemannian metric that is compatible with the observed structure exists? I believe if the objective is actually a (sufficiently smooth) distance there are existence results, but it would be nice with arguments for the more general cases in sec 3.1
-- it is assumed that a chart is known. One can always find a mapping to a lower-dimensional subspace such as is done in the MNIST example, but whether the data is actually 2d is unknown, and the uncertainty in the chart estimation is not taken into account
-- in practice, the result would likely be a Riemannian metric approximation in a chart that is estimated with high uncertainty and with strong assumptions on the dimensionality. The data would likely not be dense in higher dimensions, and so the estimated structure would very likely be a poor approximation. This can then be used for e.g. geodesic interpolation, but whether this is actually useful is not clear to me
-- what happens if the model is misspecified? The link functions are assumed to be known a priori, right?
- the developed estimation techniques have merit, but the fact that a Riemannian metric can be recovered from the metric (distances) is not a new result


**Questions:**

Convincing counterarguments to the weaknesses listed above.

- do you require the charts to be normal? If so, that restricts the set of metrics you can possibly learn, right? I.e., it could be that a true underlying Riemannian structure was not orthonormal with respect to the chosen chart

**Limitations:**

the limitations of the method could be described more precisely

---

> ### Author Rebuttal · Authors · 2023-08-10
>
> We acknowledge the existence of related literature learning Riemannian metrics in lines 22-26 and will include more as suggested by reviewers. We hope the proposed framework can shad new light on this topic. See also our general response point 3. See our general point 4 for limitations.
>
> ## usefulness of the method
> As we discussed in the general point 5, existence results for such compatible metric do exist. [Fefferman et al. 2020](https://doi.org/10.1137/19M126829X) also discussed a manifold reconstruction problem similar to our additive error case (eq. (3.2)). Their discussion focuses on abstract metric spaces with no pre-specified coordinates, while our setup is simpler as we assume that the coordinate chart is readily available.
>
> Indeed lower-dimensional subspaces are always obtainable while the question is how faithful they would be to present the original data. We consider this a modeling choice while specifying the coordinate chart as a step in, e.g., tSNE. In fact, assessing the uncertainty of the proposed models (3.1) could also potentially provide insight to the quality of dimension reduction: for example, a large uncertainty may suggest poor coordinate representations.
>
> The link function is pre-specified and the model can be misspecified. Though, our focus is the connection in (3.1) and (3.5), therefore only those simpler models (3.2) – (3.4) (as commonly seen in generalized linear models) are included. As we discussed in the general point 1, more flexible models are possible under the proposed framework, and that the effect of the link function is expected to be small.
>
> ## Do you need normal coordinates?
> No, the chart need not be normal. See line 140 of the main text. The Christoffel symbols are coordinate dependent and vanish under normal coordinates. Our method does not assume this and the Christoffel symbols are estimated.

---

> > ### Comment · Reviewer_9sCn · 2023-08-12
> >
> > Thanks for the rebuttal. My scoring has not changed.

---

> > > ### Author Response · Authors · 2023-08-16
> > >
> > > Thank you for your review of our work.

---

### Official Review · Reviewer_5uut · 2023-07-06

**Soundness:** 4 excellent
**Presentation:** 3 good
**Contribution:** 3 good
**Rating:** 7
**Confidence:** 3

**Summary:**

This paper develops a theory for estimating the Riemannian metric tensor for a given set of observations and some addition information. This additional information includes a (noisy) measure of similarity between the given points in a pairwise fashion. Examples of this information includes the geodesic distance, or a binary response about the similarity of types, or a binary response about relative comparison.

The formulation relies on a formula for the intrinsic distance between corresponding points on two geodesics shot from the same base p.  This formula approximates the said distance using the Euclidean chords, Riemannian metric tensor, and the Christoffel symbols. The first term computes the Riemannian metric between the shooting vectors and the second term accounts for the curvature. The paper seizes on this linear approximation and sets up a regression problem for estimating these matrices from the given data. In this way, it estimates the metric tensors and the Christoffel symbol at point p. The paper further develops the estimation theory (bias and variance) for the metric tensor estimation.

This theory is demonstrated using several experiments, some based on simulated data and some on real data sets. The simulated data helps validate the method for a known geometry (sphere, double spiral). The real data experiments involve learning metrics for the taxi travel times and MNIST images embedded in R^2 using tSNE.


**Strengths:**

-- The problem of learning Riemannian metric from the data is quite important and challenging.

-- The paper formulates this problem in a novel and interesting way and provides analytical insights, rather than the current systems that apply deep neural networks to such problems.

-- The paper goes on to develop estimation theory for the metric tensor. There is some interesting statistical contributions here.

**Weaknesses:**


-- The simulation experiments seem to involve manifolds with constant curvature (unless the curvature changes along the spiral). Perhaps the authors can try their method for ellipsoids or some variable curvature manifolds.

-- How does the method work if the sampled points on the manifold are sparse? I feel that the linear regression model derived here requires the manifold points to be close otherwise the errors will start piling up.

**Questions:**



-- I believe that the equation connecting geodesic distance to the Mahalanobis distance is the same kind that is used to derive Cramer-Rao bound on estimation error using the Fisher-Rao Riemannian metric. It would be interesting to make that connection if there is one.


-- What are the practical situations where the pairwise geodesic distances between points on a manifold are given, and one does not know the Riemannian metric? I understand the taxi example but for the MNIST example one has to use an embedding which is somewhat of an arbitrary choice.

**Limitations:**


The authors have not explicitly address the limitations.

---

> ### Author Rebuttal · Authors · 2023-08-10
>
> For the practical problems with pairwise distance but no Riemannian metric, we kindly refer to our general response point 2. See also our general point 4 for limitations.
>
> ## The simulation experiments seem to only involve manifolds with constant curvature.
> Our theoretical foundation allows for general non-constant curvature and this is the backbone that makes the method work for practical applications as demonstrated. The proposed method applies to manifolds with variable curvature like an ellipsoid, and we can showcase this via an ellipsoid example if accepted and time permitted.
>
> ## How does the method work if the sampled points on the manifold are sparse?
> Sufficiently dense data is indeed necessary for the proposal, which could be demanding for high-dimensional data due to the curse of dimensionality. However, data often exhibits the manifold phenomenon, namely data intrinsically lie close to a low-dimensional manifold (see also our general point 1). Thus as illustrated in the MNIST example, we can apply the proposed method after a dimension reduction step such as tSNE. The resulting representations tend to be dense since they lie in a low-dimensional space. This can substantially alleviate the curse of dimensionality, and the dense local neighborhood requirement will more likely hold true.
>
> ## Connection to Fisher-Rao metric
> When working with a statistical manifold, namely a parametric family where each point is a distribution, the Fisher-Rao metric is the canonical Riemannian metric which enjoys nice statistical interpretation. Its component matrix is the Fisher information, the inverse of which becomes the Cramer–Rao lower bound. Amari has shown that the Kullback–Leibler divergence (like a distance) is a function whose second differential produces the Riemannian metric (e.g., [Murray Rice 1993](https://books.google.com/books?id=ZBa7F9LrDrMC&pg=PA76#v=onepage&q&f=false)). Though, we are uncertain whether there would be a direct connection to our work estimating the metric tensors, while the Fisher-Rao's are based on parametric family.

---

### Official Review · Reviewer_E2so · 2023-07-07

**Soundness:** 3 good
**Presentation:** 3 good
**Contribution:** 2 fair
**Rating:** 5
**Confidence:** 4

**Summary:**

This paper proposes a method to estimate the Riemannian metric of data space when coordinate representations of each data point and some noisy similarity measurements among data points are provided. The similarity measurements of different types, such as noise-contaminated distances, similarity/dissimilarity labels, and comparative similarity labels, are probabilistically modeled as functions of intrinsic distances between data points. The squared geodesic distances are approximated to be linear to the Riemannian metric and Christoffel's symbol via Taylor's expansion, leading the metric estimation problem to be like a maximum likelihood estimation using a generalized linear model with each entry of the Riemannian metric and the Christoffel's symbol as its parameters. Asymptotic convergence rates of the bias and variance of the estimator are derived for the case where noisy distances are given as the similarity measure. Experiments are performed using some simulated data, New York taxi trip duration data, and MNIST data sets to demonstrate the proposed estimator's benefits in capturing the underlying geometry of the data space.

**Post rebuttal** I have increased my score from 4 to 5. Lingering concerns involve providing more practical applications and comparisons to other methods.

**Strengths:**

* The paper is in general well-written.
* The paper provides some interesting ideas to characterize the Riemannian geometry of data space by utilizing noisy similarity measurements among data points, such as continuous noisy distances, binary similarity labels, and binary relative comparison labels, in a unified framework.
* The proposed method to estimate the Riemannian metric is quite simple, seems original, and shows reasonable experimental results.
* This paper provides complete proof of a proposition for the local approximation of the geodesic distance and that for the asymptotic convergence analysis of the estimator.


**Weaknesses:**

* The considered problem setting seems to be quite restrictive. It would be rare that a problem provides both low-dimensional coordinate representations and meaningful intrinsic distances among data points. Obtaining each of them has been an important research topic for decades.
* The experiments need to provide practical applications of the proposed method. For example, obtaining geodesics in the experiments is usually done to demonstrate the validity of the estimated Riemannian metrics but not for further use. Suggesting more practical uses for real data based on the estimated Riemannian metric, computed geodesics, or other subsequent geometric quantities such as lengths or volumes, would significantly benefit the community.
* This paper lacks any comparison with other Riemannian metric estimation methods. A direct comparison would be possible to the method in Perrault-Joncas et al., which considers a mapping $f$ from the data submanifold embedded in the ambient space (endowed with an ambient space metric) to a coordinate chart and estimates the push-forward metric in the coordinate chart via the mapping $f$.
* Experiments in high-dimensional settings are lacking. How the method behaves according to the dimension would be valuable information. A worry here is that local regions to estimate the Riemannian metrics would increase exponentially with respect to the dimension.


**Questions:**

* When extracting the Riemannian metric from binary similarity labels or binary relative comparison labels, there may not be any ground truth Riemannian metrics. Is it then appropriate to call the proposed framework an 'estimation'?
* How can the proposed probabilistic models be justified? Distance scale may significantly affect the results according to the current choice of models in (3.3) and (3.4). Also, different options of probabilistic models would result in other Riemannian metrics.
* Is there any guarantee that the local squared distance in (3.5) is always positive when the estimation is performed based on (3.7)? This may also be related to the point that the estimated metric is not guaranteed positive-definite. I wonder if solving for the case (3.11) and (3.12) can make the estimated squared distances, i.e., $\eta_u$s, negative.
* There are no explanations for obtaining estimates for Christoffel's symbols at arbitrary points, i.e., $\hat{\Gamma}\circ \gamma(t)$, which are required for obtaining geodesics.
* Regarding the post-smoothing of the obtained Riemannian metrics explained in the supplement, the weighted averaging of Riemannian metrics seems strange. For example, we cannot simply add tangent vectors on different tangent spaces but should apply parallel transport for them to be on the same tangent space.

**[Minor comments]**
* It must be explicitly stated in the main text the location of the supplement containing the proof of each proposition.
* Since the proposed approximations and estimation methods are valid only locally, it would be better to explain how the methods can be applied globally in the main text, not in the supplement.


**Limitations:**

The authors have adequately addressed the limitations of the paper. For other possible limitations, please refer to the weaknesses and questions above.

---

> ### Author Rebuttal · Authors · 2023-08-10
>
>
> For the weakness pointed out regarding restrictive setup, application, comparison with other methods, and dimensionality, we kindly ask the reviewer to refer to our general response points 1--3. See our general response point 5 for compatibility of the metric.
>
> The reviewer is correct that the proposed method (and most if not all local polynomial based approaches) do not scale well with increased dimension. That is why this work is based on a low-dim manifold assumption as discussed in general point 1. As we demonstrated in the MNIST example, it successfully preserve the geometry of high-dim image data with low-dim spaces.
>
> ## How can the proposed probabilistic models be justified?
> Metric estimated under a constant multiple of the distances (scaling) will only differ up to multiplication of a constant. This is a special case of conformality. For example, one can follow the definition of the Christoffel symbol (line 88) to see that it is invariant to different distance scales. In other words, the different distance scales would act like unit conversion and not affect the geometry features like angles, shapes, and the geodesic curves (but its length would differ).
> Employing different probabilistic models (specifically $g$ and $Q$) may lead to different estimated metrics, but the estimated metric will change only if the derivative of the link at 0 changes by the chain rule. Like that in the generalized linear models, this is a necessary modeling component which would be determined by data type and domain knowledge. Our contribution focuses on the possibility of estimating the metric under such framework with a spread of geodesics, for which simple models as in (3.2) – (3.4) were considered.
>
> ## Positivity
> While the estimated Riemannian metric in a finite sample is not guaranteed to be positive definite, our asymptotic theory (proposition 4.1) shows that the estimated metric will be positive definite with probability tending to 1 given sufficient observations (near the point of estimation). In our experiments we rarely had estimates that were not positive definite and we did not encounter serious non-positive issues as shown in the last part of section S4.5 (page 15) and figure S4.10 of the Supplement. One could in principle adopt constraint optimization to enforce positive definiteness as that in the metric learning, but for simplicity we chose to employ an unrestricted algorithm that is more computationally efficient.
> Model (3.2) is general in that it can include both positive and negative distances, but in practice only positive distances are observed and our model is applied on these more intuitive setups. One could switch the link function and/or the loss function in a similar manner as in local quasi-likelihood [(Fan, Heckman, and Wand 1995)](https://www.tandfonline.com/doi/abs/10.1080/01621459.1995.10476496) to ensure strictly non-negative distance. But again, we chose to present the simplest model here.
>
> ## There are no explanations for obtaining estimates for Christoffel's symbols at arbitrary points.
> Following (3.6) – (3.9), the Christoffel symbols can be estimated simultaneously with the metric, and analogous post-smoothing steps (see section S3) also apply. Post-smoothing is adopted mainly to speed up the geodesic computation. So that we do not need to re-estimate the tensors at every point as requested by the ODE solver, which is time consuming. We also notice the ODE solver (and the resulting geodesic curves) benefited from the added smoothness especially for the cases with binary similarity measures.
>
> ##  The weighted averaging of Riemannian metrics seems strange in the post-smoothing step.
> Since we are working on a chart, post-smoothing is applied on the component functions instead of on the (coordinate-free) tensor directly, so comparison of tangent vectors on different tangent spaces is not involved. More specifically, under the local coordinate chart $(x^1, \dots, x^d)$, the estimated metric tensor is fully spelled out as $G_{ij} dx^i dx^j$. We post-smooth the estimate of the (continuous) component functions $G_{ij}, i, j = 1, \dots, d$. Implicitly, we assume the coordinate chart contains the data domain, so that transition maps (to bridge different local coordinate charts) are not involved.
>
> ## Minor comments
> - *It must be explicitly stated in the main text the location of the supplement containing the proof of each proposition.*
>
> We will explicitly state this in the camera-ready version if accepted. Thank you for pointing this out.
>
> - *Since the proposed approximations and estimation methods are valid only locally, it would be better to explain how the methods can be applied globally in the main text, not in the supplement.*
>
> We will add a short sketch of the post-smoothing and refer to the Supplement in the main text.

---

> > ### Comment · Reviewer_E2so · 2023-08-16
> >
> > I appreciate the author for the responses. Many questions have been answered, but there still exist some concerns as follows:
> > - Applications beyond obtaining geodesics are lacking.
> > - About justifying the probabilistic models: what bothered me was that the output of models in (3.3) and (3.4) can depend on the scale, i.e., too close distances will make Y = 0 for all data points and vice versa. This may not affect the estimation, though. Can you elaborate more on 'the estimated metric will change only if the derivative of the link at 0 changes by the chain rule'?
> > - Comparisons to other benchmarks seem required.
> > - I still think weighted averaging of the Riemannian metrics is a bit *ad hoc*. Mentioning tangent vectors was just an example. We typically do not sum Riemannian metrics defined on different points.
> >
> > Due to the above aspects, I will maintain my score for now.

---

> > > ### Author Response · Authors · 2023-08-16
> > >
> > > To respond to your additional concerns:
> > > - We also demonstrated an application in Sec 6 on estimating the underlying cost of travel through estimating the Riemannian metric.
> > > - We may now better appreciate the reviewer's question. Model (3.3) and (3.4) are postulated for all pairs or triplets of data points, and there is no specification of scale involved. For geometry, because the Riemannian metric is a local quantity, it depends on the pairwise distances only through those measured for close-by points, so in some sense only the responses for these pairs/triplets of points are important. However, this is quite different from saying that the model depends on scale because the postulation of the models is scale-independent.
> > >
> > >     Only the estimation process involves scale, namely the bandwidth $h$ in (3.10), as a tuning parameter for the bias-variance tradeoff. We take the analogy of nonparametric regression where there is a single underlying target (the conditional mean function) while the bandwidth is a device to approach a consistent estimate. Again, here the scale is only needed for the estimation.
> > >
> > >     If the distance between two locations $X_{uj}$ is too close, the response $Y$ will most likely be 0, so too small a tuning parameter $h$ will result in no variation in the neighborhood, blowing up the estimation variance. However, when the bandwidth is well chosen according to bias-variance tradeoff (namely, trading off the availability of the data and the neighborhood radius), there will be moderate variation in the response within the neighborhood, and thus the estimation will be consistent. This is also the phenomenon we saw from the numerical demonstrations.
> > >
> > >     By (4.1) and the model (3.1), we can derive that if $X_{u0}$ and $X_{u1}$ are close, $E(Y_u\mid X_{u0},X_{u1}) \approx Dg^{-1}(0) \times \delta^i_{u,0-1}\delta^j_{u,0-1}\beta_{ij}^{(1)}$, where $Dg^{-1}(0)$ denotes the derivative of $g^{-1}$ at 0. The targeted Riemannian metric will thus change only if the derivative of the link at 0 changes.
> > > - There is a lack of methods and implementation which are close in the geometric perspective to our method, even though methods for producing geodesics are quite common. For comparability issues, we did not include other methods as benchmarks.
> > > - The Riemannian metric is a smooth tensor field, so interpolating/smoothing over nearby location is well justified both theoretically and practically. Our choice is for computation consideration, and in principle, our method can be implemented without interpolating the metrics.

---

> > > > ### Comment · Reviewer_E2so · 2023-08-17
> > > > **Thank you**
> > > >
> > > > Thank you for further clarifications. I will increase my score.

---

### Author Rebuttal · Authors · 2023-08-09

We truly appreciate reviewers' careful read of our manuscript and helpful comments. We summarized common weakness/questions and provide below our general response. See our replies to individual reviews for specific responses.

## 1. Restrictive problem/model setting
*The reviewers question that our settings could be strict: a. low-dimensionality and meaningful distance, b. probabilistic formulation in eq (3.2) – (3.4), and c. a pre-specified link function.*

1. Our method and calculation are designed for data drawn from a manifold with low intrinsic dimensions. The observed similarity measures are generated based on intrinsic geodesic distance, as in eq (3.1). The low-dimensional manifold assumption ([Bengio et al. 2013](https://doi.org/10.1109/TPAMI.2013.50)) is commonly satisfied by real-world image and audio datasets because of the manifold phenomenon, and also satisfied in perception studies in psychology as pointed out by a reviewer. The manifold assumption enables us to apply nonlinear dimension reduction and avoid the curse of dimensionality, even though the raw data, like images, live in a high-dimensional ambient space.
2. The probability model is proposed as a principled guide for method development and also for theory. The specific models enlisted in Example 3.1 and Example 3.2 are given as commonly encountered scenarios that can be handled by our framework; the proposed method is not limited to these models.
3. The choices of link function $g$ and loss function $Q$ are flexible and can accommodate a variety of data generating mechanisms similar to what generalized linear model (GLM) can handle. We expect the effect of the link function to be small because the estimation is done locally and only the derivative of the link function at the origin matters.

## 2. Lack of practical application/importance
We demonstrated two applications from transportation and computer vision. We expect our method to also be widely applicable in perception studies but the authors are not familiar with open data there.

Problems do exist where pairwise distances are easily obtainable but the driving geometry is not. Other than our taxi and MNIST examples, correlations among fMRI signals (track recurrent coactivation of neurons) are used to analyze brain functional connectivity ([van den Heuvel and Hulshoff Pol 2010](https://doi.org/10.1016/j.euroneuro.2010.03.008)). Another (remotely) related topic is travel time tomography, where the internal structure of media (e.g., organ, earth) is estimated by the travel time of waves (e.g., ultrasound, seismic). They also utilize pairwise measures to capture underlying structure.

In many cases, the data space is non-Euclidean, thus a Riemannian metric is needed to capture the intrinsic geometry. Yet, finding an appropriate one can be a challenging task but of great interest, as it could lead to more accurate similarity measures, better clustering algorithms, and improved recognition systems. We refer to literature in our main text and the general point 3 and references therein.

## 3. Existing methods learning Riemannian metric
*There is a large body of existing literature learning Riemannian metric, and there is no comparison to existing methods such as that in \[18\].*

We will include additional citations to acknowledge more recent literature utilizing latent variable models, including those based on pull-back ([Arvanitidis, Hauberg, and Schölkopf 2020](http://arxiv.org/abs/2008.00565)) and/or GP-LVM ([Jørgensen and Hauberg 2021](https://proceedings.mlr.press/v139/jorgensen21a.html)).
Meanwhile, stemming from distance metric learning, our work focuses on connecting the similarity measures to the Riemannian metric and its higher order derivatives (as in our eq. (3.1) and (3.5)). Admittedly there are plenty of rooms for further development including benchmarking to comparable methods, which cannot be exhausted in a single paper.

Perrault-Joncas et al. (2013)’s method ( \[18\] ) targets a more limited scenario in that it starts with pairwise extrinsic Euclidean distances without noise, while our method handles general distance metric with noise. Their method can thus not be applied to most of the numerical demonstrations in our work except for the MNIST digit. There is no code implementation of their work available which makes it difficult for comparisons.

## 4. Lack of open discussion on limitations
We do recognize several limitations and discuss those in section S1. We appreciate the reviewers’ interest and a deep dive into our Supplement. The limitation section is put here because of the page limit for the main text. We will add a more apparent remark in the main text linking to them.

## 5. What are we estimating?
*Can we always find a Riemannian metric compatible with the similarity measure, so that we are actually “estimating” some well-defined quantities, especially for binary measures?*

It is not always guaranteed that a (distance) metric space is a Riemannian manifold, let along when the measurements are binary. Analogously, in order for an underlying Riemannian metric to exist, the similarity measure needs to be induced by the geodesic distance on some Riemannian manifold. Thus our method is based on the manifold assumption (Bengio et al. 2013) and that the observed similarity measures are induced by some Riemannian metric, for which we are estimating. We are not aware of a certain answer for the binary situation. However, an abstract metric space can be approximated by a Riemannian manifold (theorem 1 of \[7\]). A later paper ([Fefferman, et al. 2020](https://doi.org/10.1137/19M126829X)) also discussed a manifold reconstruction problem similar to our additive error case (eq. (3.2)).

## 6. Positivity
*The estimated metric tensors might fail to be positive-definite and model (3.2) allows negative distance.*

Due to length limit of a single comment, see the Positivity section of our response to reviewers E2so and uGdp (identical).

---

### Decision · Program_Chairs · 2023-09-21

**Decision:**

Accept (poster)

**Comment:**

This paper tackles the estimation of Riemannian metrics on latent spaces from noisy observations of distance data, either through the distances themselves or weaker observations such as relative distance differences. This problem is important and challenging, and the submitted paper presents a novel and (in this context) simple way to estimate the underlying Riemannian metric. This is valuable!

While two reviewers are enthusiastic about the paper, the other two are borderline, and during rebuttal, the authors have provided a number of clarifications, as well as some promises of further experiments. Their final version should make sure to include the clarifications and promised additional experiments, as well as an explicit and open discussion of the paper's limitations.